# Feature selection techniques for modeling tower fatigue loads of a wind turbine with neural networks

Artur Movsessian[1], Marcel Schedat[2], Torsten Faber[2]

[1]Institute for Infrastructure and Environment, School of Engineering, University of Edinburgh, United Kingdom
[2]Wind Energy Technology Institute, University of Applied Sciences Flensburg, Flensburg, 24943, Germany

*Correspondence to*: Marcel Schedat (marcel.schedat@hs-flensburg.de)

**Abstract.** The rapid development of the wind industry in recent decades and the establishment of this technology as a mature and cost-competitive alternative have stressed the need for sophisticated maintenance and monitoring methods. Structural health monitoring has risen as a diagnosis strategy to detect damage or failures in wind turbine structures with the help of measuring sensors. The amount of data recorded by the structural health monitoring system can potentially be used to obtain knowledge about the condition and remaining lifetime of wind turbines. Machine learning techniques provide the opportunity to extract this information, thereby improving the reliability and cost-effectiveness of the wind industry as well. This paper demonstrates the modeling of damage equivalent loads of the fore-aft bending moments of a wind turbine tower highlighting the advantage of using the neighborhood component analysis. This feature selection technique is compared to common dimension reduction/feature selection techniques such as correlation analysis, stepwise regression, or principal component analysis. For this study, recordings of data were gathered during approximately 11 months, pre-processed, and filtered by different operational modes, namely standstill, partial load, and full load. The results indicate that all feature selection techniques were able to maintain high accuracy when trained with artificial neural networks. The neighborhood component analysis yields the lowest number of features required while maintaining the interpretability with an absolute mean squared error of around 0.07 % for full load. Finally, the applicability of the resulting model for predicting loads in the wind turbine is tested by reducing the amount of data used for training by 50 %. This analysis shows that the predictive model can be used for continuous monitoring of loads in the tower of the wind turbine.

## 1 Introduction

Wind power is becoming the electricity-generating technology with the lowest costs in several areas of the world (REN21, 2018). To ensure the cost-competitiveness of this technology in the future, it is important to seize the potential cost reductions related to operation and maintenance (O&M). This includes improving monitoring solutions and life extension strategies. The possibility of monitoring with sensors has enabled the gathering and supervision of data regarding the condition of a structure to, e.g. detect failures. Particularly, structural health monitoring (SHM) in wind turbines (WTs) allows monitoring the structural behaviour and stresses of structures such as blades, towers, and foundations.

While machine learning techniques are widely applied in industries such as the automotive, information technology, and communication, the wind industry is starting to explore the suitability of these promising methods for their benefits. Although data-driven attempts have been made to estimate the loads acting on the turbine using available information from the SCADA system, there is no consensus yet on the type of relationship existent between these data and actual load measurements. In the last years, the focus on this topic increased. This section aims to review available scientific literature regarding modeling loads with existing SCADA data for WTs.

SHM systems could be used to verify structural safety and determine the remaining useful lifetime (RUL) of WTs (Schedat et al., 2016). Moreover, information gathered through SHM during the lifetime of WTs can potentially be used to identify structural weaknesses and feed this information back to the manufacturers, ultimately improving the design of new turbines (Ziegler et al., 2018). Another potential benefit of SHM is a decrease in maintenance costs. Typically, operation and maintenance costs (including both fixed and variable costs) represent approximately 20 to 25 % of the levelized cost of electricity (LCOE) (IRENA, 2015). SHM could reduce this share by allowing the implementation and establishment of more efficient maintenance practices such as predictive maintenance while enabling better spare-parts inventory management. Consequently, downtime is reduced and production is increased.

Currently, the assessment and evaluation of the structural condition of WTs without a load measurement system can be challenging. Particularly the estimation of fatigue loads can be difficult due to a lack of information (Melsheimer et al., 2015; Schedat and Faber, 2017). Therefore, exploring the ways to mine data from SHM systems and extract valuable information becomes an interesting and high demanded field of research.

Ziegler et al. (2018) performed a literature review and assessed the development of the lifetime-extension market of onshore WTs. The alternative of extending the lifetime of a WT, as opposed to repowering or decommissioning, is appealing given the potential increase of returns on investments (ROIs), however, not much public research has been done on this matter. The authors contributed, then, by comparing updated load simulations and inspections for lifetime extension assessments in Germany, Spain, Denmark, and the United Kingdom. For lifetime extension to be a feasible alternative, the structural integrity of the turbine should not compromise the level of safety. In this regard, the survey performed by the authors determined that, beyond the use of SCADA systems, no short-term load measurements or monitoring are carried out in the countries surveyed (a few exceptions were identified in the UK, where load reassessment is performed). They found that most interviewees focus on practical assessments for cost reasons. Nevertheless, these practical inspections are no guarantee that the safety level can be maintained during the lifetime extension. The authors concluded that new operation and maintenance strategies and data-processing methodologies are necessary for lifetime extension purposes. In this regard, data-driven approaches may contribute to the cost reduction of lifetime extension assessments.

In line with the findings of Ziegler et al. (2018), other authors have worked on the aforementioned data-driven approaches. Noppe et al. (2018), for example, reconstructed the thrust loads history of a WT based on both simulated and measured SCADA data. The data gathered corresponded to operational 1 s and 10 min. Moreover, the data is segregated into different operational modes. The selection of explanatory variables that the authors performed was based on a Pearson correlation analysis. The

first two weeks of operational data were used to model the thrust loads using neural networks and validated by one year of data. The model has the following input features: wind speed, blade pitch angle, rotor speed, and generated power. The results of this study showed that the constructed model was able to estimate thrust loads with a relative error that does not exceed 15 %. The authors also concluded that the use of simulated data yielded slightly better results and that adjustments in the hyperparameters of the neural networks had no significant impact on the estimated thrust loads.

Relatedly, Vera-Tudela and Kühn (2014) focused on the selection of variables to be used for fatigue load monitoring and attempted to define an optimum set of explanatory variables for that purpose. The authors identified 117 potential variables (13 statistics of 9 SCADA signals) used in related scientific literature. Among them, the mean of generator speed, electrical power, and pitch angle have been the most commonly used. The authors decided to apply several feature selection methods to six sets of variables. The methods chosen included Spearman coefficients, stepwise regression, cross-correlation, hierarchical clustering, and principal components. To evaluate the outcomes of the feature selection methods a feedforward neural network was employed. The authors concluded that principal components yielded the best set of variables, however, the resulting set lost expertise knowledge about the relation between the variables. In this sense, ranking the variables by their corresponding Spearman coefficients resulted in a fair compromise between the number of features required to monitor the damage equivalent load for blade out of plane bending moment and the available expert knowledge.

Smolka and Cheng (2013) examined the amount and type of data necessary to determine a fatigue estimator for the operational lifetime of a WT. The inputs for the neural network are selected through a correlation analysis applied to standard data statistics of available SCADA signals such as electrical power, generator speed, pitch angle, among others. The authors concluded that the minimum training data sample size required is approximately half a month worth of measurements.

Seifert et al. (2017), acknowledging the complexity and cost of handling extra measurements, assessed the minimum needed size of a training sample to predict fatigue loads using 10 min statistics of SCADA signals and neural networks. In a sense, Seifert et al.'s work is an extension or continuation of Vera-Tudela and Kühn's (2014) and Smolka et al.'s (2013). Seifert et al. (2017) tested different sample sizes varying between one day (i.e., 144 records) and four months (i.e., 4032 records) of measurements. They determined that a sample of 2016 records of 10 min statistics are sufficient to predict flap wise blade root bending moments of a WT independent of seasonal effects.

The reconstruction or estimation of loads using statistics from SCADA data was already presented and tested in the mid-2000s. Cosack and Kühn (2006) developed a stepwise regression model for estimating the rotor thrust. Despite the good results (i.e. deviations between the calculated and the estimated loads ranging from 5.4 % to 7.3 % in the worst case), the presented model was too complex and time-consuming with further restrictions. In a new development of the model, an estimation method for the corresponding target values (i.e. damage-equivalent loads and load magnitude distributions) used neural networks (Cosack, 2010; Cosack and Kühn, 2007).

The performance of artificial neural networks depends on the quality of the information provided to them, thus, the features used to train them are key to obtaining high accuracy in the results with a parsimonious model. So far, little research has been done regarding feature selection for modeling tower fatigue loads. The available literature has focused on techniques such as

correlation analysis, principal component analysis (PCA), and stepwise regression to select the best subset of information. This paper aims to contribute to this body of literature by assessing the use of Neighbourhood Component Analysis (NCA) as a feature selection technique to extract relevant information from SCADA data to train artificial neural networks and model fatigue loads.

The paper is organized as follows: Section 2 outlines the methodology followed in this study, section 3 summarizes the results, and, finally, section 4 presents the conclusions derived from the obtained results.

## 2   Data and Methodology

### 2.1   Wind turbine and SCADA data

This paper seeks to model the tower fatigue loads of a commercial wind turbine with a rated power of 2.05 MW, a hub height
of 100 m, and a rotor diameter of 92.5 m located in the northern part of Germany. The turbine is used for research purposes by the Wind Energy Technology Institute at the Flensburg University of Applied Sciences. For this study, the readings from the SCADA and a load measurement system in the previously mentioned turbine were recorded for around 11 months and collected in 10 min files. The tower bottom bending moment is measured by strain gauges. These were installed and wired as full-bridge (Wheatstone) with temperature compensation. A Wheatstone bridge is widely used in strain gauge applications
because of its ability to measure small deviations in resistance. The calibration factors were determined from the results of the shunt-resistor-calibration, tower geometry, and the thickness of the tower wall at the strain gauge positions (provided by the turbine manufacturer). The offsets are determined through a yaw round. The sensors used to extract features for the model are described in Table 1 and were selected based on a literature review and consultations with an application engineer.

**Table 1 – Description of SCADA sensors selected**

| Feature name | Description | Unit of measurement | Frequency [Hz] |
|---|---|---|---|
| **Explanatory variables** | | | |
| Omega | Rotational speed at the rotor | rpm | 20 |
| acc_x | Acceleration fore-aft (x-direction) | mm s$^{-2}$ | 20 |
| acc_y | Acceleration side-side (y-direction) | mm s$^{-2}$ | 20 |
| v_wind | Wind speed | m s$^{-1}$ | 20 |
| v_dir | Relative wind direction | degree | 10 |
| omega_gen | Rotational speed at the generator | rpm | 20 |
| air_density | Air density | kg m$^{-3}$ | 20 |
| Pitch | Pitch angle | degree | 20 |
| ACpow | Active power output | kW | 20 |

| Feature name | Description | Unit of measurement | Frequency [Hz] |
|---|---|---|---|
| **Dependent variables** | | | |
| Bieg1_060_240 | Bending moment derived from a gauge sensor located at 60° & 240° inside the tower bottom | kNm | 50 |
| Bieg2_150_330 | Bending moment derived from a gauge sensor located at 150° & 330° inside the tower bottom | kNm | 50 |

The strain gauge measurements at the turbine were transformed into a resultant fore-aft tower bending moment, which was later used to calculate the short-term damage equivalent load (DEL) for every 10 min time series. This calculation was performed through a rainflow counting algorithm and, later, the resulting load spectrum was further reduced to a constant load range. After several equivalent cycles, this load range results in the same equivalent accumulated damage as the spectrum of loads previously calculated through the rainflow counting algorithm. The short-term DELs were calculated following Equation (1):

$$S_0 = \left[ \frac{\sum_i n_i * S_i^m}{n_{eq}} \right]^{\frac{1}{m}} \tag{1}$$

where $n_{eq}$ is the equivalent number of cycles, $S_i$ the different load ranges, $n_i$ the corresponding cycle numbers and $m$ is given by the slope of the Stress-Cycle (S-N) curve of the material used for the tower (DNV/Risø, 2002). In this case, it is assumed an inverse slope $m = 3$ for the steel tower and $n_{eq} = 9.5064$ for the 10 min time series (equivalent to $10^7$ cycles in 20 years). The DELs were then used as the dependent variable of the model.

## 2.2 Methods

The methodology followed in this study is graphically described in Figure 1. First, the sensors which provide relevant information to model resultant fore-aft tower bending moments were selected (see Table 1). In the next step, the resulting records were analyzed for missing data (e.g. zero values) and outliers as there were periods where the turbine was out of service or measurement failures with no registered data. Subsequently, affected records were removed. The process of outlier detection in this study was not automated but done through visual inspection of the descriptive statistics calculated from the time series for each operational mode. To determine the relationship between the dependent and nine explanatory variables described previously, each of the 10 min files was summarized by estimating the following descriptive statistics for every explanatory variable: i) minimum value, ii) maximum value, iii) arithmetic mean, iv) range, v) mode, vi) standard deviation, and vii) variance.

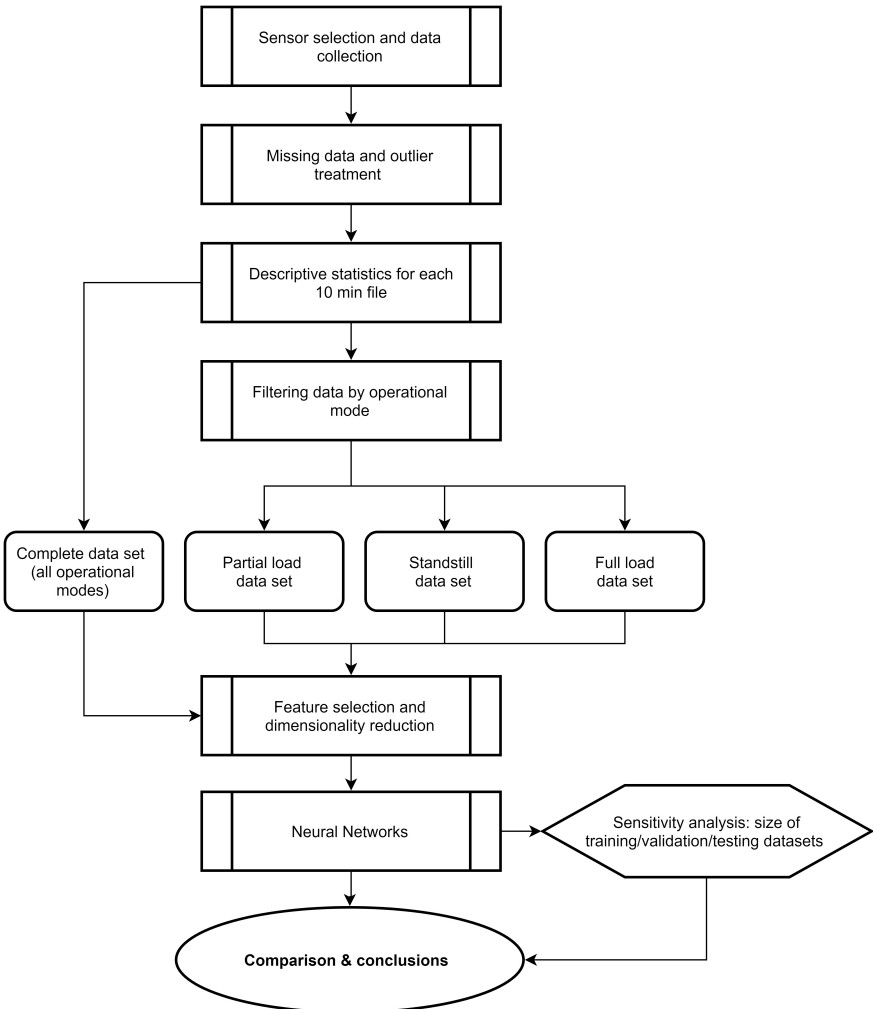

**Figure 1 – Main methodological steps**

In this way, the dataset consists of 63 features (i.e. explanatory variables, see appendix A 1). Excluding the time where no SCADA data were recorded, the total amount of data results in 36266 (77.1 %) observations. This corresponds to a little over

5    eight months of useful information.

Relationships between sensor signals and the estimated DELs can vary depending on the operational mode of the wind turbine, e.g. the pitch angle operates mainly during startup and full load. Methods with an underlying linear assumption, such as the correlation analysis, can lead to misinterpretation of feature importance when observing the complete dataset. Therefore, the data were filtered by operational modes, namely standstill, partial and full load to study the relevance of the different potential

10   features for these operational modes. Additionally, this filter enables the construction of individual models to account for the particularities of each operational mode, thereby improving the accuracy of the monitoring system. The filtering was done employing the feature "ACpow" which refers to active power output. In this sense, standstill corresponds to 10 min mean

"ACpow" readings below or equal to 5 kW (0.25 % of nominal power); partial load to readings higher than 5 kW and below or equal to 2000 kW (97.56 % of nominal power); and full load to readings above 2000 kW.

Research by Sharma and Saroha (2015) concluded that a reduction of dimensions possibly leads to a better performance of the mining algorithms while maintaining a good accuracy, therefore, it is important to eliminate potential redundant data and select

5 the variables with most predictive power for the model. For this, three different feature selection techniques and one dimension reduction technique were applied to the entire dataset and the datasets resulting from filtering the data by operational mode. These techniques include Pearson correlation, stepwise regression, NCA, and PCA. Pearson correlation measures the linear correlation between two variables and maps the result to an interval between -1 and 1, where 0 indicates no linear relationship (Boslaugh and Watters, 2008). It can be calculated as per Eq. (2):

$$r_{XY} = \frac{\sum_{i=1}^{n}(X_i - \bar{X})(Y_i - \bar{Y})}{\sqrt{\sum_{i=1}^{n}(X_i - \bar{X})^2}\sqrt{\sum_{i=1}^{n}(Y_i - \bar{Y})^2}} \tag{2}$$

where $n$ is the sample size, $X_i$ and $Y_i$ are the observations with index $i$ and $\bar{X}$ represents the arithmetic mean of all samples. A threshold value of 0.5 was set to define the strength of the correlation. In this sense, a correlation coefficient between 0 and 0.49 is weak and a correlation coefficient between 0.5 and 0.95 is strong. Correlation among all features of a particular sensor above 0.95 was considered as a redundant sensor and, therefore, eliminated from further analyses.

Stepwise regression is an iterative method where features are added and removed from a multilinear model based on their statistical significance in the regression (Draper and Smith, 1998). The algorithm begins by constructing an initial model with one feature (forward selection) or all the features (backward selection) and continues adding or removing features by comparing the explanatory power of the larger or smaller models. At each step, the p-value of the corresponding F-statistic is estimated and compared to a threshold p-value to decide which features are included in or excluded from the model. P-value

is used as a probability measure to identify if a particular feature is significant for the outcome of the model. If a p-value is larger than 0.05 the null hypothesis is true and the feature is selected for further modeling. The algorithm repeats this process until the added feature does not improve the model or until all features that do not improve the explanatory power of the model are removed. This method is considered to be locally optimal, yet not globally optimal given that the selection of features included in the initial model is subjective and there is no guarantee that a different initial model will not lead to a better fit.

NCA is a non-parametric classification model used for metric learning and linear dimensionality reduction (Goldberger et al., 2005). It is based on a modeling technique known as k-Nearest Neighbours (k-NN), which is a supervised learning algorithm used for classification or regressions (Han and Kamber, 2006; Parsian, 2015). In its simplest form, the k-NN approach looks for the closest k = 1 observation to the query observation $x_q$ within the training dataset by measuring the distances to the neighbouring data points and selecting the one that satisfies $\min_i$ distance $(x_i, x_q)$. The output is then predicted by applying a

function y = h(x) where h is the trained k-NN prediction function. In a multidimensional dataset, the k-NN approach requires to differentiate between the "relevance" of the explanatory variables for the intended output. For the learning process, different weights can be assigned to the features of the model using the "Scales Euclidian Distance" estimation detailed in Eq. (3):

$$Distance(x_i, x_q) = \sqrt{a_1 \big( x_i[1] - x_q[1] \big)^2 + \cdots + a_d \big( x_i[d] - x_q[d] \big)^2} \qquad (3)$$

where $x_i$ is a vector of input values, $x_q$ is the query vector, $a$ is the scaling number that defines the relevance of each explanatory value, and $d$ the total number of features. The weights are assigned randomly and then adjusted by solving a minimization problem (minimizing the prediction error). Other distance metrics can be used, namely Mahalanobis, Manhattan, rank-based, correlation-based, and Hamming (Hazewinkel, 1994).

Lastly, PCA is a statistical method to reduce the dimensions of a dataset that presumably contains a large number of irrelevant features while retaining the maximum information possible (Vidal et al., 2016). This is done by transforming the original set of multidimensional data into a new set referred to as components employing eigenvectors and eigenvalues. A pair of eigenvector and eigenvalue indicate respectively the direction and how much variance is there in the data in that direction. The eigenvector with the highest eigenvalue is the first principal component. In this sense, the transformation allows reducing the dimensions of the dataset to a few components with relatively low loss of information.

Table 2 summarizes the strengths and limitations of all the methods considered for feature selection and dimension reduction in this study.

**Table 2 – Comparison of strengths and limitations of methods used**

| Method | Description | Strengths | Limitations |
|---|---|---|---|
| **Pearson correlation** | Measures of the strength of a linear association between two variables | - Measures the degree and direction of correlation between the variables through a coefficient<br>- Widely used and easily interpretable<br>- Computationally inexpensive | - Supervised<br>- Affected by extreme values in the data<br>- Assumes a linear relationship between variables<br>- Prone to misinterpretation in case of homogeneous data |
| **Principal component analysis** | Dimensionality reduction technique through linear transformation | - Unsupervised<br>- Well-established technique<br>- Reduces overfitting<br>- Reduces redundancy of a feature set given the orthogonal components | - Assumes that the principal components are a linear combination of the features<br>- Low interpretability<br>- Uses variance as the measure of importance<br>- Prone to loss of information as high variance axes are treated as principal components, while low variance axes are treated as noise |
| **Stepwise regression** | Step-by-step iterative construction of a regression model that involves the selection of independent variables to be used in a final model | - Able to manage large amounts of potential predictors<br>- Easily interpretable and tractable<br>- Computationally inexpensive | - Supervised<br>- Sensitive to collinearity<br>- Highly dependent on the order in which features are added or removed to the model |
| **Neighborhood component analysis** | Feature weighting approach which optimizes the nearest neighbour classifier performance to address the issue of high dimensionality of the training data | - Rarely leads to overfitting due to cross-validation<br>- Non-parametric<br>- Performance does not degrade as training data size increases | - Supervised<br>- Usually necessary to select a value of the regularization parameter<br>- Sensitive to the choice of loss function |

In this way, 16 neural networks (NN) were developed for four datasets (all operational modes, standstill, partial load, and full load), three feature selection techniques, and one dimension reduction technique. Each dataset is divided into training, validation, and testing subsets. For this, 70 % of a dataset is randomly chosen and used by NN for training the model, 15 % are used for testing, and 15 % for validation, i.e., this subset is used to adjust the model through the mean squared error (MSE).

This adjustment stops when the MSE does not significantly improve. The validation subset is used as a measure to avoid overfitting the NN and generalize the prediction model. After that, the model can be applied to new datasets. The test subset does not affect training or validation, it is only used to measure the performance of the trained NN.

The NN models used in this paper are trained with the Neural Network Toolbox from MATLAB (MathWorks, 2019). The standard settings consist of a two-layer feed-forward NN with a sigmoid transfer function in the hidden layer and a linear

transfer function in the output layer. The NN was initially set to 25 neurons in the hidden layer and one neuron in the output layer as per Lind et al. (2017). However, we tested different configurations and found that the results remain consistent. Therefore, the number of neurons in the hidden layer is set to 10 neurons and one neuron in the output layer. This simple configuration reduces the computational complexity and time while enabling the modelling of non-linear relationships. The Levenberg-Marquardt algorithm is selected as the training algorithm. The results from the 16 models were compared to derive

conclusions about the relationship between operational data and tower loads acting on WTs.

Finally, the predictive capability of the model for continuous monitoring is tested. For this purpose, the NN is trained using only the first 50 % of the data gathered during partial load. The prediction error is estimated to determine the accuracy of the model.

## 3    Results and discussion

This section describes the sensors identified by different methods as potential predictors of tower fatigue loads of the WT and presents the results of using a predictive model for continuous monitoring.

### 3.1    Feature selection and dimension reduction

Before building a model to predict the desired output, it is important to define which variables could act as predictors. The feature selection methods described in Sect. 2.2 were applied to four different datasets: i) an eight months dataset, ii) a full-

25 load dataset, iii) a partial load dataset, and iv) a standstill dataset. The results of the feature selection methods are described below. For detailed information on selected features for each operational mode, the reader is referred to appendix A 1.

#### 3.1.1    Complete dataset: eight months data

A Pearson correlation analysis was applied to the pre-selected features for predicting the DELs of the fore-aft bending moment of the tower. Before using the features with the strongest correlation in a model, it is necessary to check for collinearity, i.e.,

the correlation between independent variables. A high correlation between two explanatory variables suggests that these

variables should be excluded from the model to avoid collinearity issues. From this analysis, it was determined that rotational speed at the rotor should be excluded from the model and only rotational speed at the generator should be included given that these two features are a factor away from each other and, thus, may add bias to the model due to redundancy. This resulted in 56 features from the initial 63 (see Sect. 2.2).

The correlation analysis shows that only 27 of the 56 features are strongly correlated and should be used as independent variables in the model. The accelerations in both directions (i.e., x- and y-axis) are highly correlated with the DELs. The standard deviation of the acceleration in the x-direction presents the highest correlation with a coefficient of 0.97, depicting an almost linear relationship between this feature and the dependent variable. Additionally, several statistics of wind speed and power output are also highly correlated with the DELs. As an example, the bilinear relationship between the mean wind

speed and the calculated DEL (Fig. 2) is graphically shown over the complete measurement campaign and for all operational modes in Fig. 2c below.

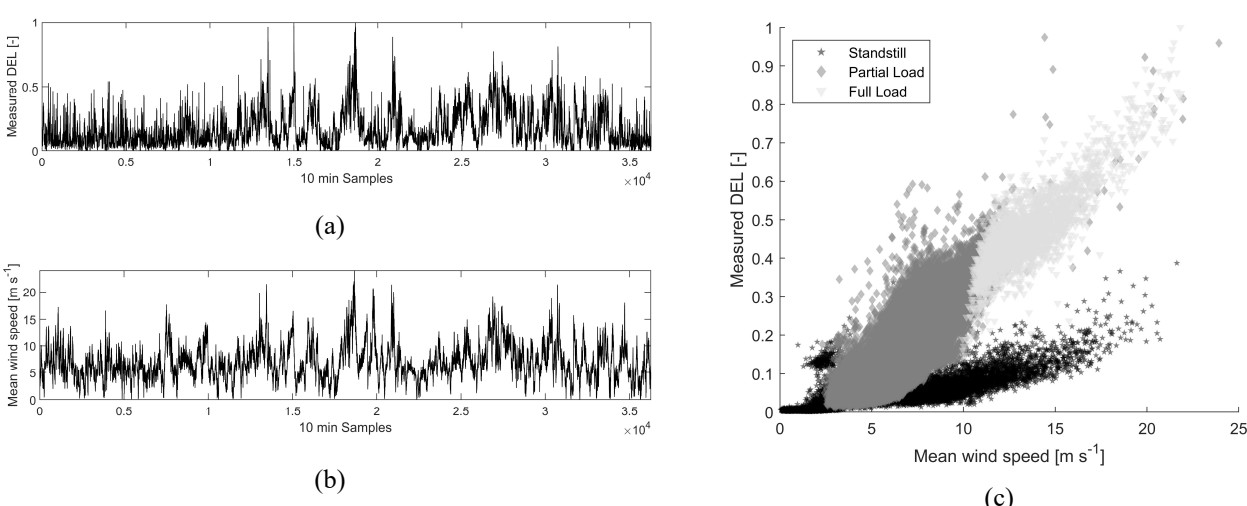

**Figure 2 – Time series corresponding to (a) normalized measured DELs, (b) mean wind speed and (c) scatterplot of both mean wind speed and measured DELs with a correlation coefficient of 0.74 when data is not filtered by operational modes.**

These results suggest that these features fluctuate together with the dependent variable and they could potentially be used to build a model that can estimate the DELs of the fore-aft bending moments of the tower without installing strain gauge sensors. Furthermore, the results show that air density and relative wind direction, along with all their corresponding descriptive

statistics, have a very low correlation with the DELs and should be, therefore, disregarded in the model based on this feature-selection technique. The mean wind direction, in particular, has a correlation coefficient close to zero, indicating an

insignificant linear relationship with the DELs. Many of the remaining variables are highly correlated with each other, nevertheless, they add potentially valuable information to the model.

An alternative would be the use of a method such as PCA which could contribute to avoiding multicollinearity by transforming the data while maintaining the information contained in it. After using PCA on the remaining 27 features, 12 Principal Components are identified and can be used to build a model. This data was transformed as explained in Sect. 2.2 estimating the variance explained by each of the first components as seen in Figure 3. It can be observed that 99 % of the information contained in the features is now stored in the first 12 components. The remaining 15 components explain less than 1 % of the cumulative variance. A model could be built using the first 12 components and the results should be almost as accurate as using the 27 features selected after the correlation analysis. The biggest disadvantage with this method is that given the transformation of the data, it is no longer possible to interpret it. The results, nevertheless, remain interpretable and are free of the influence of multicollinearity.

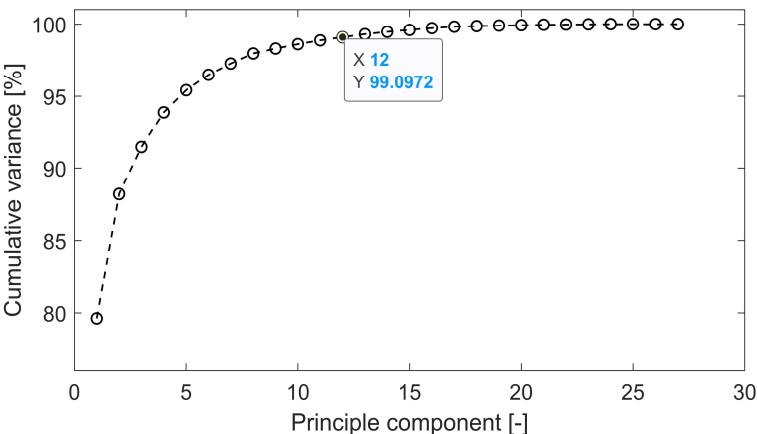

Figure 3 – Cumulative variance explained by principal components

Alternatively, an interactive stepwise regression was built using the pre-selected 56 features. Different combinations of features were tested to identify those that should not be included in the model given that they do not contribute to the predictive power or result in an increase in the error of the model. The features with a p-value above 0.1 should be omitted from the model. The results suggest excluding a total of 30 variables from the regression model. Among these can be found the minimum, maximum, mean, and range of the rotational speed at the generator, most descriptive statistics of air density, except for the standard deviation, and range, mode, and standard deviation of the acceleration in y-direction.

It is important to highlight that the variable that represents the range of the acceleration sensor in the x-direction was identified as statistically insignificant despite its high correlation with DELs. As mentioned earlier, the possible models explored with the stepwise regression are limited. The algorithm builds different models from the 56 features depending on the order in which these features are added to (in the case of forward selection) or removed from (in the case of backward elimination) the models. In this sense, the range of the sensor "acc_x" and the variance of "acc_x", which are correlated with a factor of 0.90, could be

considered mutually exclusive. The decision as to which of these variables to include in the model would depend solely on which variable is added or removed first in the stepwise regression. In this case, the algorithm suggests excluding the range of "acc_x", a highly correlated feature, based on the search for the local minimum instead of evaluating all combinations. Ultimately, this method identified 33 features as statistically significant and, thus, these are included in the model.

5 The last feature selection method, NCA, was applied as well to the dataset with 56 features. 13 features were identified by this method as relevant for the prediction of DELs, a significantly smaller number than those selected by applying the correlation analysis and stepwise regression.

To summarize, mean values and standard deviations are the descriptive statistics that can best describe the data according to the three feature selection methods applied. Features selected by all three methods include wind speed, acceleration, and power 10 output.

### 3.1.2 Data filtered by operational modes

The dataset was divided by operational modes into 10 min samples with 7825 (21.6 %) of the measurements corresponding to standstill, 25604 (70.6 %) to partial load, and 2837 (7.8 %) to full load. Each dataset contains 56 features and the corresponding DELs.

15 The first feature selection method used in these new datasets is again the Pearson correlation analysis. The results show that most of the descriptive statistics for wind speed and acceleration are highly correlated with the DELs in all operational modes. The first differences appear in the generator speed. As expected, the generator speed is not relevant during standstill since the rotor is not moving or is only idling. The mean generator speed is not as relevant in full load as it is in partial load. During full load, the rotational speed is around a specified number and must be kept as stable as possible. Therefore, the mean rotational 20 speed does not change significantly during full load. During partial load, the mean rotational speed is within a higher range, therefore, it has a higher correlation with the corresponding DELs.

During full load, the standard deviation and variance of the rotational speed are highly correlated with the DELs. The standard deviation explains how the values differ from the mean, thus, conclusions about the dynamics of the turbine can be derived based on these spreads. For example, fluctuations of the rotational speed during full load have a significant effect on the tower 25 movement which explains the high correlation between the standard deviations of the rotational speed with the DELs. Additionally, several descriptive statistics of the pitch angle are correlated with the DEL exhibiting correlation coefficients greater than 0.5. This correlation is only significant during full load. The pitch angle is held at the most efficient lift-to-drag ratio during the partial load and, therefore, not many variations can be observed during standstill and partial load. During full load, the turbine pitches continuously to keep the rotational speed nearly constant. For each operational mode, PCA was 30 performed to account for potential collinearity in the feature-set. This was done consistently with an explained variance of 99 % remaining.

The second feature selection method applied is stepwise regression. The results are not consistent with the correlation analysis. Air density and wind direction did not correlate with the DELs, however, they were chosen by the stepwise regression during

standstill and partial load as potential predictors. Also, the pitch was chosen as a significant variable during standstill, even though the turbine is not pitching. In general, the modeler needs to be careful when interpreting the results from a stepwise regression as described in Sect. 2.2.

NCA was applied to the three datasets. Examining the results, one significant difference to the correlation analysis is that the wind direction was identified as significant during standstill and partial load by the NCA, whereas the correlation analysis showed no correlation of these features with the output during any operational mode. Furthermore, the range of the pitch angle was identified as relevant during partial load, which was not the case in the correlation analysis. Mean acceleration in the x-direction and the standard deviation were the only two features identified as significant by the NCA in all three operational modes.

## 3.2 Modeling fatigue loads

Once the features with predictive power have been identified for the different datasets, NN are built to evaluate the predictions. The outcomes of these models are described hereafter.

### 3.2.1 Eight-month data with all operational modes

The first analysis is conducted on eight-month data without filtering by operational modes. To illustrate the results, features used for training the NN are selected by using the correlation analysis and can be examined in appendix A 1. The data is randomly split into training, testing, and validation sets. The regression model of the NN in Figure 4 shows a similar R-value in all regressions indicating that there is no overfitting in the model.

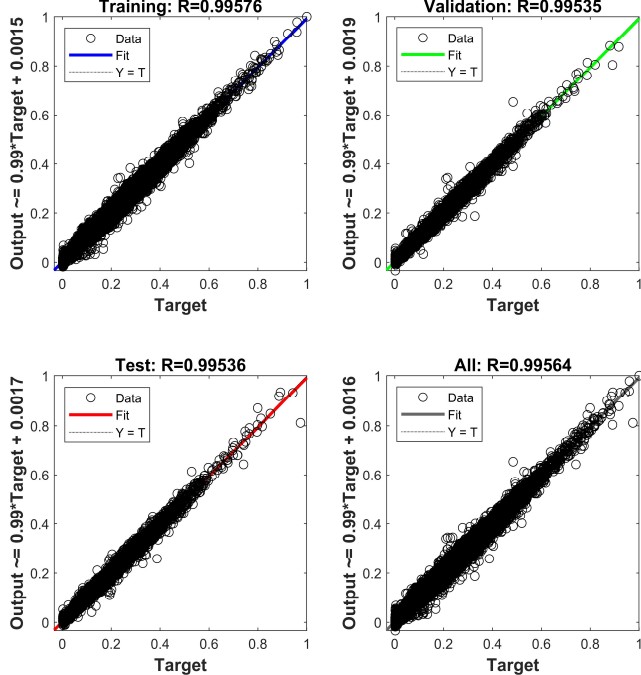

**Figure 4 – Linear regressions between the normalized neural networks prediction and the measured DELs. The NN are built using the complete data set (i.e., eight months) and 27 features selected after the Pearson correlation analysis.**

The R-value for the complete data-set is 0.99564 which can be confirmed by observing the top plot in Figure 5 where the predicted DELs overlap with the measured DELs with a mean prediction error of 2.22 % (see Table 3). Nevertheless, the prediction error, shown in the bottom plot in Figure 5, can be as high as 685.79 %. Values close to zero can have a significant impact in terms of the mean error in percent due to a high ratio of prediction and measured DELs.

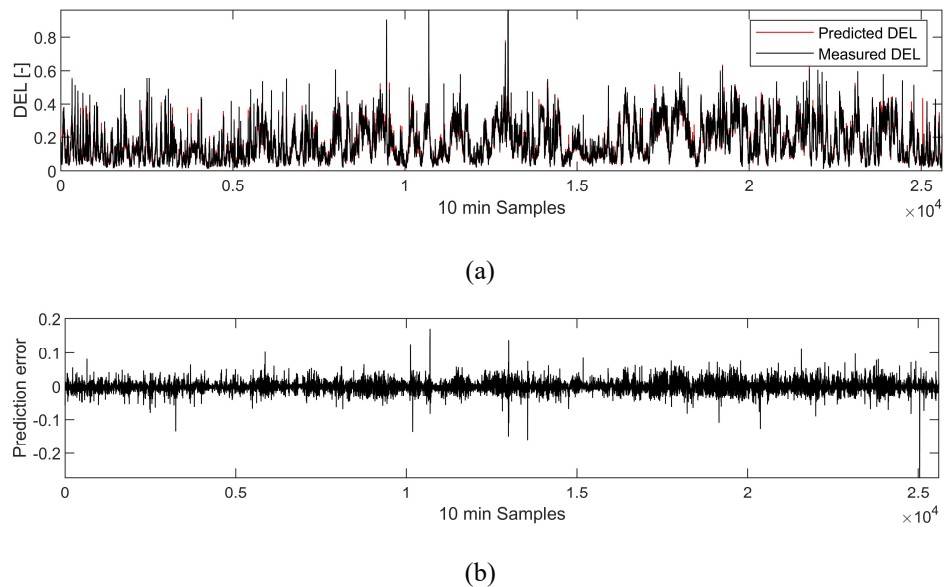

(a)

(b)

**Figure 5 – NN prediction: Plot (a) presents the normalized predicted and the measured DELs by NN. Plot (b) is the prediction error.**

5    Plotting the error against the wind speed for all operational modes in Figure 6, it can be concluded that the mean prediction error is significantly higher at low wind speeds. If the wind speed is below approximately 3 m s$^{-1}$, the WT is at a standstill.

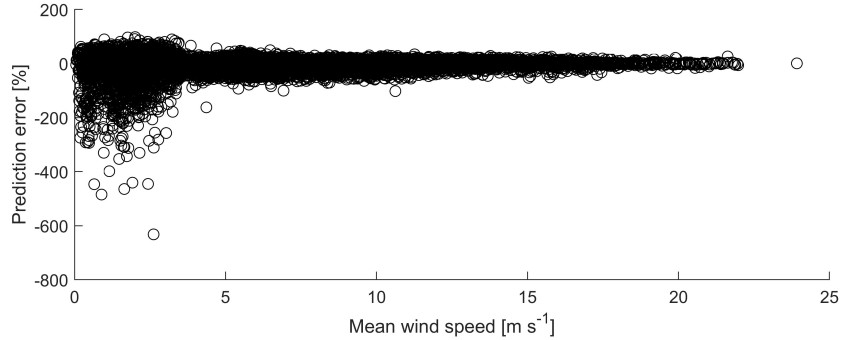

**Figure 6 – Wind speed against prediction error**

Comparing the results in Table 3, it can be seen that the model using features selected by NCA results in the lowest mean error.

10   Overall, these results are significantly higher with the mean error ranging from 2.07 to 2.94 % than those obtained by Vera-Tudela and Kühn (2014) with the mean error ranging from 0.01 to 0.22 %. This can be explained by the high prediction error during low wind speeds seen in Figure 6. However, our results indicate that it is possible to significantly reduce the number of features used in the model by applying NCA while maintaining a low prediction error. NCA did not select features such as the

variance of the acceleration in y-direction and the minimum windspeed during a 10 min time series, which had been selected by the correlation analysis and stepwise regression. This shows that to model the DELs with NN, these features are not relevant and can be omitted without compromising the model's performance as suggested by the mean error of 2.07 % in Table 3. Furthermore, the non-parametric nature of NCA enabled this technique to outperform the other techniques such as correlation and stepwise regression which are based on the assumption of linear relationships.

**Table 3 – Summary of results from Neural Networks for the complete one-year considering all operational modes**

| N° | Feature subset | No. of features (% of total) | R | | | Mean error [%] | Std dev [%] | Max abs error [%] | Mean abs. Error [kNm] |
|---|---|---|---|---|---|---|---|---|---|
| | | | Training | Validation | Test | | | | |
| 1 | Correlation | 27 | 0.99576 | 0.99535 | 0.99536 | 2.22 | 22.85 | 685.79 | 237 |
| 2 | Correlation & PCA | 12 | 0.99393 | 0.99400 | 0.99365 | 2.94 | 18.93 | 410.33 | 276 |
| 3 | Stepwise | 33 | 0.99555 | 0.99523 | 0.99476 | 2.24 | 7.23 | 671.48 | 224 |
| 4 | NCA | 13 | 0.99581 | 0.99593 | 0.99568 | 2.07 | 26.09 | 525.20 | 228 |

### 3.2.2 Data filtered by operational modes

This section explores the performance of the NN when built with data subsets from different operational modes. It can be observed that the mean percent error is significantly high in standstill compared to other operational modes as shown in Table 4. Nevertheless, the mean absolute error in kNm is the lowest. The high maximum error observed previously when using the complete eight-month dataset (i.e., when using "all operational modes") for the different training sets could be explained by the poor predictive power of the data from the standstill mode. When the NN was built using filtered data for partial and full load, the errors of the predictions decreased significantly. Thus, it can be concluded that the data from the standstill mode adds uncertainty to the model. This can be observed in Figure 7.

Presumably, the small variations observed in the readings from the sensors during standstill do not provide enough information to predict the DELs. This is consistent with the R-values of the model during standstill mode. These values are the lowest among the different operational modes. A more detailed look at this case would be necessary to derive valuable insight.

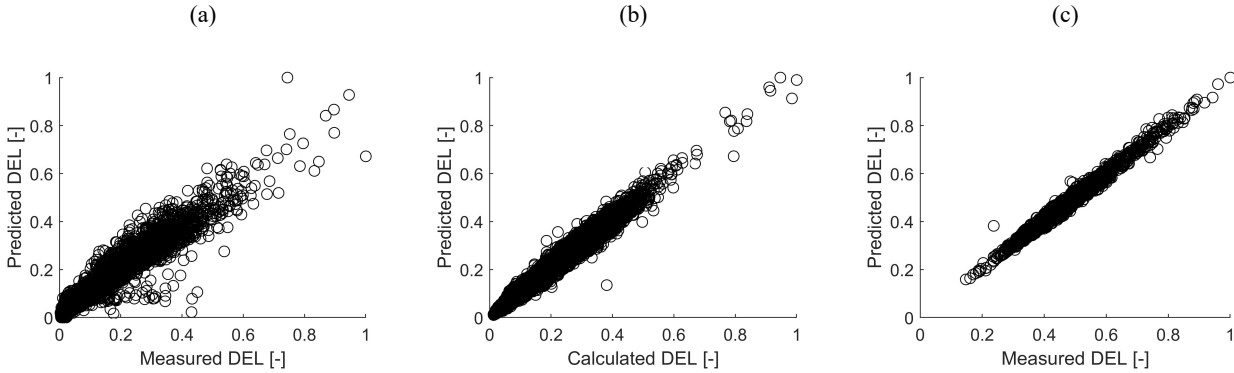

(a)                                    (b)                                    (c)

**Figure 7** – **Normalized measured vs. predicted DELs. Figures correspond to a model built with features from the correlation analysis. Subfigure (a): Standstill. Subfigure (b): Partial load. Subfigure (c): Full load.**

Moreover, Table 4 shows that the partial and full-load models constructed using smaller sets of features derived from the application of methods such as PCA or NCA have approximately the same predictive power as those models constructed using larger sets of features derived from applying methods such as stepwise regression or correlation analysis. This can be observed in the comparison of the measures of goodness-of-fit (i.e. R-values) among the models. Nevertheless, it is important to mention
5    that to apply PCA the complete feature-set is needed to transform all the information in the first few components. This is not the case when applying NCA, where the most relevant features are directly identified.

The application of feature-selection and dimension-reduction methods can be considered a good practice. NCA outperformed all other methods in terms of the mean error in standstill and full load. In partial load, NCA still performed well, however, correlation and correlation & PCA yielded slightly lower mean errors.
10   The results can be compared to the existing work from Vera-Tudela and Kühn (2014). In the case of the full load model, the mean error, the maximum absolute error, and standard deviation of the error are in similar ranges. However, the accuracy of the results from the partial load model is slightly worse for all features-sets.

**Table 4 – Summary of results from Neural Networks for different operational modes**

| N° | Subset | No. of features | R Training | R Valida-tion | R Test | Mean error [%] | Std dev [%] | Max abs error [%] | Mean abs. Error [kNm] |
|----|--------|-----------------|------------|---------------|--------|----------------|-------------|-------------------|-----------------------|
| | | | | *Standstill* | | | | | |
| 1.1 | Correlation | 18 | 0.96884 | 0.96598 | 0.96962 | 9.89 | 37.46 | 472.83 | 189 |
| 1.2 | Correlation & PCA | 7 | 0.96211 | 0.96776 | 0.95255 | 14.13 | 40.01 | 432.56 | 199 |
| 1.3 | Stepwise | 35 | 0.98781 | 0.98352 | 0.97053 | 7.21 | 46.51 | 828.72 | 138 |
| 1.4 | NCA | 16 | 0.98369 | 0.98163 | 0.97891 | 6.60 | 39.77 | 505.83 | 145 |
| | | | | *Partial Load* | | | | | |

| | | | | | | | | | |
|---|---|---|---|---|---|---|---|---|---|
| 2.1 | Correlation | 28 | 0.99322 | 0.99228 | 0.99212 | 0.70 | 8.77 | 82.34 | 242 |
| 2.2 | Correlation & PCA | 12 | 0.99045 | 0.99034 | 0.99040 | 0.69 | 9.74 | 89.06 | 256 |
| 2.3 | Stepwise | 37 | 0.99330 | 0.99295 | 0.99217 | 1.21 | 9.03 | 71.16 | 239 |
| 2.4 | NCA | 11 | 0.99282 | 0.99236 | 0.99218 | 0.72 | 9.33 | 79.02 | 240 |
| | | | | ***Full Load*** | | | | | |
| 3.1 | Correlation | 28 | 0.99186 | 0.98858 | 0.98389 | 0.06 | 3.07 | 56.28 | 276 |
| 3.2 | Correlation & PCA | 12 | 0.98759 | 0.98568 | 0.98636 | 0.11 | 3.58 | 53.35 | 290 |
| 3.3 | Stepwise | 23 | 0.99175 | 0.99011 | 0.98953 | 0.07 | 2.79 | 16.29 | 272 |
| 3.4 | NCA | 8 | 0.99048 | 0.98847 | 0.99084 | 0.07 | 3.09 | 54.85 | 273 |

### 3.2.3 Continuous monitoring with a predictive model

In this section, the results of using a predictive model for continuous monitoring are presented. The aim is to identify if and how the errors in the outcomes of the model vary when using only the first 50 % of data gathered. The model is tested by predicting the DELs corresponding to the remaining share of the data. The majority (i.e., 70.6 %) of all data gathered corresponds to the partial load mode, therefore, this subset was selected for this analysis. As in the previous analysis, four models are built using the feature-sets from the correlation analysis, correlation and PCA, stepwise regression, and NCA.

Figure 8 shows the prediction error from the model using the feature-set from the correlation analysis. It can be seen that the mean absolute prediction error from the model trained using the first 50 % of the data gathered is 211 kNm (see Table 5). This value is lower than the mean absolute error from using this trained model to predict the remaining 50 % of the data, which yields 244 kNm (see Table 6). Nonetheless, the contrary is true for the mean error (in percentage). This variation can be explained by the same relationship observed previously in Figure 6 and Fig. 7 where the prediction error decreases at high wind speeds. As can be seen in Figure 9, the average mean wind speed is higher in the second half of the partial load dataset, which was used for testing.

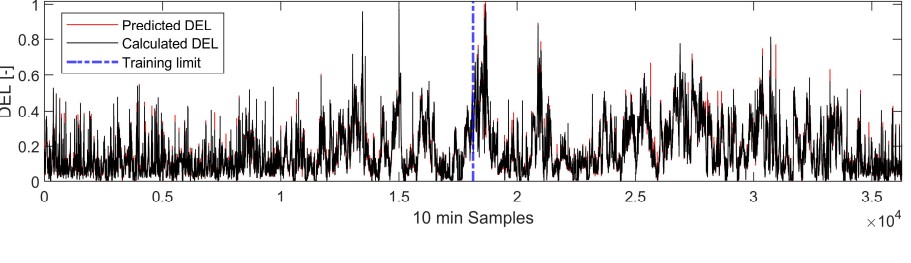

(a)

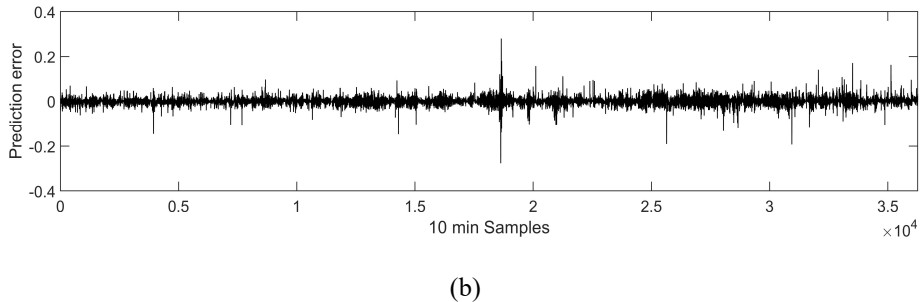

(b)

**Figure 8 – (a) A comparison of normalized predicted and measured DELs and (b) the corresponding prediction error for the model using the feature-set from the correlation analysis.**

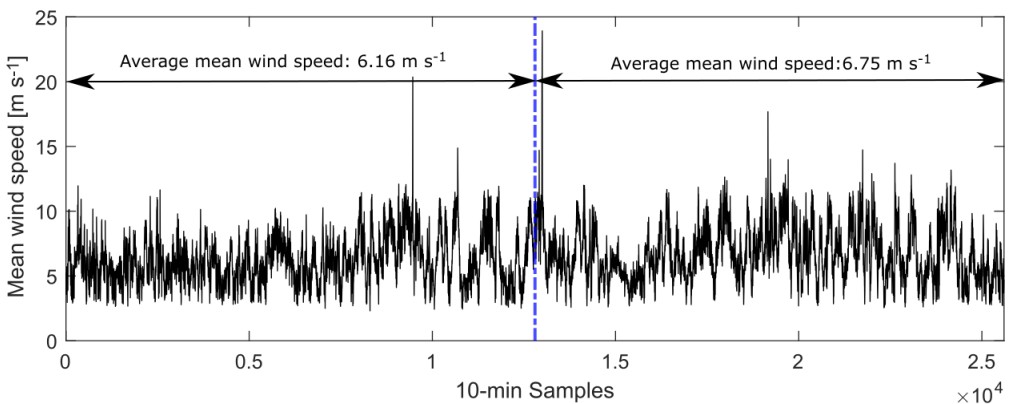

**Figure 9 – Mean wind speed during partial load**

The same behavior is observed in the remaining models. Overall, the mean error is lower in the results from the models trained using the second half of the datasets as can be seen when comparing Table 5 and Table 6.

**Table 5 – Summary of results of the partial load model trained with the first 50 % of the data**

| N° | Feature subset | No. of features (% of total) | R | | | Mean error [%] | Std dev [%] | Max abs error [%] | Mean abs. Error [kNm] |
|---|---|---|---|---|---|---|---|---|---|
| | | | *Training* | *Validation* | *Test* | | | | |
| 1 | Correlation | 27 (48%) | 0.99276 | 0.99011 | 0.99182 | 0.92 | 9.60 | 76.44 | 211 |
| 2 | Correlation & PCA | 9 (16%) | 0.99393 | 0.99400 | 0.99365 | 1.44 | 11.03 | 94.23 | 229 |

| N° | | No. of features (% of total) | | | | | | | |
|---|---|---|---|---|---|---|---|---|---|
| 3 | Stepwise | 38 (68%) | 0.99555 | 0.99523 | 0.99476 | 2.45 | 11.24 | 75.56 | 231 |
| 4 | NCA | 13 (23%) | 0.99581 | 0.99593 | 0.99568 | 1.32 | 10.21 | 75.09 | 213 |

**Table 6 – Results from using the trained model to predict the remaining 50 % of the data**

| N° | Feature subset | No. of features (% of total) | Mean error [%] | Std dev [%] | Max abs error [%] | Mean abs. Error [kNm] |
|---|---|---|---|---|---|---|
| 1 | Correlation | 27 (48%) | 0.19 | 8.98 | 60.86 | 244 |
| 2 | Correlation & PCA | 9 (16%) | 0.07 | 9.54 | 83.67 | 229 |
| 3 | Stepwise | 38 (68%) | 0.84 | 9.55 | 88.64 | 248 |
| 4 | NCA | 13 (23%) | 0.46 | 8.81 | 96.51 | 234 |

## 4    Conclusions

This paper used available SCADA data as well as strain gauge measurements from a research WT to develop a predictive model to estimate the DELs of the fore-aft bending moments of a WT tower. The dataset included over eight months of useful data. Different feature selection methods and a dimension reduction technique were applied to choose the sensors with the strongest predictive power. The data were then inputted into a feedforward neural network. The methodology and data used reproduces and enhances the approaches of similar studies in the field of SHM.

The results indicate that using all data and applying NCA for feature selection yields an interpretable and low dimensional feature-set while maintaining high accuracy. Additionally, dimension reduction techniques such as PCA can contribute to a more parsimonious model reducing the number of features needed, however, compromising the interpretability of the inputs given the transformation of the data.

The results were significantly better, i.e., yielded lower mean absolute errors, when the dataset was divided by operational modes. The models were significantly more accurate when analyzing the operation of the turbine at full load and partial load. The outcome of the model using signals from when the turbine was standing still was rather inaccurate with mean errors ranging from 6 % to 14 %. In partial load, the errors vary between 0.69 % to 1.21 % and in full load between 0.07 % to 0.11 %. It can be concluded that the performance of NN is influenced by the operational mode of the WT.

Finally, a model for continuous monitoring was built. For this, the first 50 % of partial load data was used for training and shows stable results in terms of prediction accuracy for the remaining data. All feature selection techniques showed similar results when predicting DELs for continuous monitoring. The feature set resulting from the application of correlation analysis

and PCA yielded the lowest mean error, yet the second largest standard deviation for these errors. Since the results are not significantly different for each feature selection technique, the use of NCA is preferred for the following reasons:

- A significant reduction of features (up to 86 % during full load), which also leads to faster modeling of the DELs
- The interpretability of features is maintained.

This study showed that NCA can be included as a reliable and efficient feature selection method for modelling tower fatigue loads with NN, particularly due to its non-parametric nature. Nevertheless, the performance of this technique relative to other techniques such as correlation analysis or stepwise regression will depend on the particularities of the case study (operational conditions, availability and location of the sensors, characteristics of the WT, etc.). The decision of which technique should be used to build the NN model should be based on the knowledge of the strengths and limitations of the techniques in

consideration.

This study was limited to only one WT. To be able to generalize the results obtained from this study, the NN model requires validation with data collected from a different wind turbine with the same specifications. By doing this, it will be possible to determine the relationship between SCADA data and fatigue loads with more precision, thereby eliminating the need to install expensive gauge sensors to estimate these loads and contributing to more efficient SHM methods.

Furthermore, the methodology developed during this study could be further tested through an analytical aero-elastic model. Such a model would provide larger datasets for standstill and full load to test the predictive capabilities for continuous monitoring without the significant costs that this would imply if done empirically. The results of the NN trained with information from the aero-elastic model can be compared to the results presented in this paper to derive conclusions on the reliability and accuracy of this methodology. Finally, the results could benefit from exploring alternative machine learning

algorithms such as support vector machine and k-Nearest Neighbors.

## 5    Appendix A

*A 1 Feature selected by the different methods and for the different operational modes*
*Note:The numbers correspond to the Pearson correlation coefficient.*
*The small letters next to the coefficients indicate that the feature has been selected by the method: **a** corresponds to correlation*
*analysis, **b** to stepwise regression and **c** to NCA.*

| | Number | Feature | Standstill | Partial Load | Full Load | All Modes |
|---|---|---|---|---|---|---|
| *Acceleration fore-aft (x-direction)* | 1 | acc_x_min | -0.09 | 0.06, b | -0.05 | -0.16, b |
| | 2 | acc_x_max | 0.92, a, b, c | 0.92, a, b | 0.88, a, b | 0.96, a, b |
| | 3 | acc_x_mean | 0.91, a, b, c | 0.93, a, b, c | 0.96, a, b, c | 0.96, a, b, c |
| | 4 | acc_x_range | 0.92, a, c | 0.92, a | 0.88, a | 0.96, a |
| | 5 | acc_x_mode | 0.53, a, b | 0.08, b | 0.25, b | 0.19, b |

| | Number | Feature | Standstill | Partial Load | Full Load | All Modes |
|---|---|---|---|---|---|---|
| | 6 | acc_x_std | 0.93, a, b, c | 0.94, a, b, c | 0.97, a, b, c | 0.97, a, b, c |
| | 7 | acc_x_var | 0.77, a, b | 0.85, a, b | 0.94, a, b | 0.86, a, b |
| Acceleration side-side (y-direction) | 8 | acc_y_min | 0.04 | 0.04 | 0.03 | -0.08 |
| | 9 | acc_y_max | 0.81, a, b, c | 0.90, a | 0.82, a | 0.86, a, b, c |
| | 10 | acc_y_mean | 0.80, a, b | 0.88, a, b | 0.81, a, b | 0.85, a, b |
| | 11 | acc_y_range | 0.81, a, b, c | 0.90, a | 0.82, a | 0.86, a, c |
| | 12 | acc_y_mode | 0.35 | 0.14 | 0.00, b | 0.06 |
| | 13 | acc_y_std | 0.80, a, c | 0.90, a, b | 0.83, a | 0.85, a |
| | 14 | acc_y_var | 0.70, a, b | 0.77, a, b | 0.82, a | 0.75, a, b |
| Wind speed | 15 | v_wind_min | 0.65, a | 0.53, a, b | 0.59, a, b | 0.64, a, b |
| | 16 | v_wind_max | 0.76, a, c | 0.90, a | 0.87, a | 0.80, a |
| | 17 | v_wind_mean | 0.74, a, b | 0.83, a, b, c | 0.84, a, b | 0.74, a, b, c |
| | 18 | v_wind_range | 0.75, a | 0.92, a, b | 0.77, a | 0.79, a |
| | 19 | v_wind_mode | 0.72, a, b | 0.80, a | 0.78, a, b | 0.71, a, b |
| | 20 | v_wind_std | 0.73, a, b | 0.94, a, b, c | 0.81, a | 0.77, a, b, c |
| | 21 | v_wind_var | 0.70, a, b | 0.90, a, b | 0.77, a, b | 0.71, a, c |
| Relative wind direction | 22 | v_dir_min | 0.12, b | 0.08 | 0.12 | 0.16 |
| | 23 | v_dir_max | -0.13, b | -0.05, b | 0.12 | -0.16 |
| | 24 | v_dir_mean | -0.02, b, c | 0.05, b | 0.03 | 0.00, b, c |
| | 25 | v_dir_range | -0.15 | -0.07 | 0.01 | -0.19, b, c |
| | 26 | v_dir_mode | 0.02, c | 0.00 | 0.00 | -0.01 |
| | 27 | v_dir_std | -0.15, b, c | 0.07, b, c | 0.11, b | -0.11, b |
| | 28 | v_dir_var | -0.11, b | 0.04, b | 0.10 | -0.08, b |
| Rotational speed at the generator | 29 | omega_gen_min | -0.08, b, c | 0.55, a, b | -0.79, a, b | 0.64, a |
| | 30 | omega_gen_max | 0.19, c | 0.80, a, b, c | 0.81, a | 0.68, a |
| | 31 | omega_gen_mean | 0.04, b | 0.71, a, b | 0.08 | 0.67, a |
| | 32 | omega_gen_range | 0.38, b, c | 0.37, c | 0.88, a | 0.28, c |
| | 33 | omega_gen_mode | 0.02, b | 0.65, a, b | -0.05 | 0.66, a, b |
| | 34 | omega_gen_std | 0.33, b | 0.29 | 0.94, a, b, c | 0.19, b |
| | 35 | omega_gen_var | 0.30, b | 0.22, b | 0.93, a, b | 0.13, b |
| Air density | 36 | air_density_min | -0.10 | 0.22 | 0.05, b, c | 0.23 |
| | 37 | air_density_max | -0.11 | 0.22 | 0.06 | 0.23 |
| | 38 | air_density_mean | -0.10 | 0.22 | 0.05 | 0.23 |
| | 39 | air_density_range | -0.02, b | 0.01 | 0.08 | -0.07 |
| | 40 | air_density_mode | -0.10 | 0.22 | 0.05, b | 0.23 |
| | 41 | air_density_std | -0.02, b | 0.00, b | 0.09 | -0.07, b |
| | 42 | air_density_var | -0.02, b | 0.02, b | 0.07 | -0.03 |

| | Number | Feature | Standstill | Partial Load | Full Load | All Modes |
|---|---|---|---|---|---|---|
| *Pitch angle* | 43 | pitch_min | 0.27, b | 0.03, b | 0.71, a, b, c | -0.35, b |
| | 44 | pitch_max | 0.41 | 0.31, b | 0.87, a | -0.25, b |
| | 45 | pitch_mean | 0.35 | 0.21, b | 0.81, a, b, c | -0.31, b |
| | 46 | pitch_range | 0.30 | 0.31, c | 0.55, a | 0.37 |
| | 47 | pitch_mode | 0.36, b | 0.12, b | 0.66, a, b | -0.32, b |
| | 48 | pitch_std | 0.30 | 0.24, b | 0.32, b, c | 0.25, b |
| | 49 | pitch_var | 0.26, b | 0.15, b | 0.28 | 0.09, b |
| *Active power output* | 50 | ACpow_min | -0.15 | 0.64, a, b, c | -0.05 | 0.82, a, b |
| | 51 | ACpow_max | 0.20, b | 0.89, a, b | 0.83, a, b | 0.89, a, b |
| | 52 | ACpow_mean | 0.04, b | 0.81, a, b, c | 0.29, b | 0.88, a, b, c |
| | 53 | ACpow_range | 0.21, c | 0.92, a, c | 0.17 | 0.70, a, c |
| | 54 | ACpow_mode | 0.00, b | 0.75, a, b | -0.15 | 0.85, a, b |
| | 55 | ACpow_std | 0.21, b, c | 0.88, a, b | -0.07, c | 0.59, a, b, c |
| | 56 | ACpow_var | 0.20, b | 0.70, a, b | -0.12 | 0.45, b |

## 6    Data availability

The high-frequency measurements from the SCADA and strain gauge sensors are not available due to confidentiality issues.

## 7    Author contributions

The work was carried out by AM, based on his Master Thesis at the Wind Energy Technology Institute under the supervision of MS and TF. AM preprocessed the data for machine learning purposes, implemented feature selection techniques, modeled fatigue loads with neural networks, and ran a sensitivity analysis. MS initiated the issue, ran the data-gathering campaign, and processed the raw data. All authors were involved in the development of the manuscript.

## 8    Competing interests

The authors declare that they have no conflict of interest.

## 9    Acknowledgments

We acknowledge financial support by Land Schleswig-Holstein within the funding program Open Access-Publikationsfonds.

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
