# Peer review of "Modelling tower fatigue loads of a wind turbine using data mining techniques on SCADA data"

_Wind Energy Science, 2019_

## Referee Comment (RC1) · Anonymous Referee #1 · 31 Jan 2020

As indicated in the contribution, the work presented in this paper originates from the Master's Thesis of the first author. Based on this contribution, this seems to be very good work for a Master's Thesis. In my opinion however, the quality should be improved for this paper. More in particular, especially the presentation quality. Nevertheless, the work presented is interesting and valuable.

Throughout the comments that follow, I indicated the most important (and thus necessary to tackle) ones by (*).

**General comments**

(*) In general, my main concern is the lack of shown results. Without more information, it is hard to review the results and conclusions drawn. I'll give specific examples later on. A general rule of thumb could be: whenever a conclusion is drawn based on own results, it can only be checked if the results themselves are shown. So all main conclusions drawn in the paper should be preceded by the results shown in a figure or table.

Some other general comments:

- The title states that "data mining techniques" (plural) are used to model fatigue loads. However, only neural networks are used. I'd consider to use "neural networks" instead of "data mining techniques". If you want to express the comparison of multiple techniques, an adjustment of the title might be considered towards feature selection (since this is not reflected in the title at this point). In my opinion, this is not strictly necessary.
- Although mentioned in the abstract, the sensitivity analysis regarding the length of the data set (and the motivation for it) is not mentioned in the introduction.

**Specific comments**

- P.1 line 19-20: it's not clear to me what you mean by "more conservative models regarding the number of features"
- P.1 abstract: consider including quantitative results (e.g. errors between measured and estimated DEL do not exceed x%)
- (*) P.4 line 9-10: Some more information on the turbine would be appreciated, in particular rated power.
- (*) P.5 line 1: More information about the measurement setup is needed. E.g. Are the measurements corrected for wall temperature before calculating bending moments? Is it possible to show the resulting (normalized) bending moments? E.g. vs time (for a smaller period) and vs windspeed
- P.5 line 9-11: the number used as $n\_eq$ is usually given too
- P.5 line 14-16: How are the outliers detected? What were the (normalized) limit values to detect them?
- P.5 line 19: Can you give some more explanation on the descriptive statistic "mode"?
- P.6 Figure 1: nice figure, does help to understand the methodology
- p.6 line 4: This seems to be a lot of missing SCADA data. Are all of the variables missing or is it mostly due to one variable?
- P.6 line 5: This sentence is a bit confusing. The filtering by operational modes is done for the feature selection and the sensitivity analysis, isn't it?

- (*) P.6 line 6-8: should the mean value of ACpow be below or equal to 5kW for the datapoint to correspond to standstill? Or is it the minimum, maximum or another descriptive statistic?
- P.6 line 7-8: It is easier to interpret the limits as x% of rated power instead of x kW. Consider to change to relative values.
- P.7 line 7-8: The sentence "Correlation coefficients above 0.95 …" deserves more explanation. Does this mean that if an explanatory feature correlates with more 0.95 to the bending moment it is considered as redundant? Or is this only the case for explanatory features among each other?
- P.7, line 12: more explanation about p-value and F-statistic would be appreciated.
- P.7 line 22-23 "the output is then predicted by applying a function": What kind of function?
- P.7 line 27: how are the weights decided for this paper?
- P.7 line 28: observations = features ? This is a bit confusing, since "observations" might also be used for different measurements (in time)
- (*) Section 3.1: a visualization or overview of the selected and disregarded features for each dataset and technique is missing. As a reader, it is impossible to know which features were selected by which technique for which dataset except for the (few) mentioned in the text. Only by showing these results, the drawn conclusions can be checked. Moreover, it's easier to understand the conclusions if you can see the results yourself.
- P.9 line 19: a figure showing results about the collinearity would be helpful for this discussion
- P.9 line 22-23 "Many of the remaining variables … model": an example with specific resulting values would be helpful
- P.10 line 6: How does stepwise regression avoid multicollinearity?
- P.11 Section 3.1.2: Discussion about PCA seems to be missing
- (*) P.12 Section 3.2.1: I like the idea to first give more precise results for one neural network model before comparing all of them. However, more information and results should be given. Which were the final features used for this one model? How did the training, testing and validation data look like (for example plot of the measured normalized power curve for all three datasets, can be based on mean statistics)? Plots of the measured and predicted DEL vs mean windspeed, errors vs mean windspeed for example.
- P.12 Section 3.2.1: some of the information and results given seem irrelevant for this discussion. E.g. at which epoch the training stopped. Consider to omit this from the paper or to include it in the discussion to show the added value
- P.12 Figure 3: Personally, I don't think this plot adds value to the paper. If not needed for main conclusions, consider to remove it.
- P.12 line 11-13: these datapoints (outliers) are not visible in the figure. Maybe consider a different visualization
- P.12 line 10-11 and p.13 Figure 4: Personally I don't see the added value of this figure and discussion.
- (*) P.12 line 12 and further: didn't you exclude outliers from your dataset? Did you take a closer look to the time signals of bending moment and different SCADA parameters to check what is causing these high errors?
- P.12 line 16: I cannot find the result 0.99486 in the figures or tables
- P.14 Figure 5: It's not clear to which data sets the figures exactly correspond. Add sublabels please.

- Section 3.2.1: An additional conclusion could be made: From the introduction I understood Vera-Tudela and Kühn did a similar analysis with slightly different techniques. How do your results compare to theirs?
- P.14 Section 3.2.2: Additional figures might be helpful here too. For example, measured and predicted (by the different models) DEL vs mean windspeed, where a different color is used for each model/operational mode.
- P.14 Figure 6: if you want to show DELs are lower for standstill than during full load, it is much easier to plot them on the same graph. Moreover, are the results shown here all test data? Why don't you show all test data instead of only 100 points for each operational mode?
- P.16, Section 3.3: What is the exact intention of this analysis? Is it to determine the minimum period needed to measure the bending moments? If that's the case, shouldn't the focus be on the dataset containing the least data? To make sure a good estimation is obtained for that operational mode too?
- P.16 line 11: is the dataset increased consecutive in time or by randomly picking data from the entire dataset of one year?
- Section 3.3: An additional conclusion could be made: From the introduction I understood Smolka and Cheng did a similar analysi. How do your results compare to theirs?
- (*) p.17 line 11-12: The first part of this conclusion is not clear to me. What do you mean with "conservative model regarding the number of features"? The second part doesn't seem to be true. Looking at Tables 2 and 3, the lowest mean errors are rarely found for NCA.
- P.18 line 6-10: If the purpose is to eliminate the need for installing strain gauges on every turbine, it seems it is especially necessary the model is validated on a different turbine. Training the model with data from multiple turbines might not be necessary.
- (*) P.18 line 12: Why would you want to train the neural network with larger datasets? Wasn't one of your conclusions that you didn't need as much data to have a model equally accurate?

**Technical corrections**

The comments hereafter are not critical and are meant to improve the readability of the paper.

- p.1 line 12-15: very long sentence, consider to split it up
- p.3 line 1-9: different lengths of datasets were used for the different analyses shown in this paper. The summary written here seems to suggest 2,5 months of data was used to model the thrust loads. However, only 2 weeks were used to train the model and (in case of operational data) one year was used to validate. On the other hand, 2,5 months of data was used to perform a Pearson correlation analysis.
- P.3 line 26-32: If I understood it correctly, your work is actually similar to the work of Seifert et al, except you have done it for tower bending moments, while Seifert et al. did it for blade root bending moments.
- P.6 line 4: in my opinion, 6044 hours is not easier to interpret than 36266 observations of 10 minutes. I think "a little over 8 months" would be better. Similarly in the conclusion (p.18 line 3-5)
- P.6 line 4: Considered to add also the percentage of remaining data after the removal of missing data.
- P.8 line 7: typo "squared"

- P.9 line 25: It might be helpful to clearly state that from this point the second technique, PCA, is discussed.
- P.10, Figure 2: add a line at 99% to increase visibility
- P.11 line 8: typo "datasets is  again the"
- P.13 line 5: sentence can be missed very easily
- P.14 line 10: reference to results is missing
- P.16 line 14: an equivalent number in time (weeks, months) is easier to interpret than 10241 observations

---

## Referee Comment (RC2) · Anonymous Referee #2 · 10 Feb 2020

The paper deals with the estimation of damage equivalent tower bending moments from SCADA data using a neural network approach. In particular it focuses on methods for feature selection to determine which of the available parameters has to be included as input to the network. The work is interesting and worth publication in general. However, improvements are necessary. The paper should also be revised carefully to improve the presentation quality.

General comments

The title implies that several data mining techniques for modeling tower fatigue loads are compared. However, it seems that the main work focuses on feature reduction

techniques. In my opinion the title should be adjusted to better reflect the content of the paper.

A comprehensive literature survey has been undertaken to investigate state of research related to the estimation of loads from SCADA data. However, the paper also focuses on methods for feature reduction. Are there studies outside of wind energy which compare feature reduction methods? If so, how do the results compare to those presented in this paper?

Specific comments

p.1, l.18-20: What is meant by "conservative" models?

p.1, l.22: What do you mean with "deployment"? It may not be the correct word. How does that relate to the competition?

p.2,l.7: Could you please explain why load measurement systems are required for this purpose? The structural condition is usually monitored with accelerometers.

p.4,Table 1: From the rest of the paper it seems that the dependent variables for the modeling are the fore-aft bending moment and not strains (see p.5, l.11). Please clarify.

p.5, l.2: Please explain what you mean by "short-term" equivalent load?

p.5, l.2: Please explain what you mean by "short-term" equivalent load?

p.5, l.11: Why m=3?

p.7: It is nice that the four approaches are explained briefly. However, it should also be discussed how the approaches compare with respect to feature reduction. What are their limitations or advantages? Maybe the methods can also be compared using a table which may be easier to follow for the reader?

p.8, l.8: How does the validation subset generalize the transfer function? And what is meant be "transfer function" anyhow? Is that the network itself?

p.8, l.13: The choice for the NN architecture should be based on some rational and not on a default suggestion by a software. Why do you think that the chosen NN architecture is reasonable? Can you give another justification than that this is the default setting of a software?

p.8, l.15: See comment above.

p.9, l.14: The DELs are calculated for the bending moments and not for the thrust.

p.9, l.15: What do you mean by "facing the wind"? Are measurement with large yaw errors included in the dataset? And how can the wind not affect the thrust load?

p.9, l.23ff: Is PCA not applied? Or does the rest of the page refer to PCA?

p.10,l.19-21: The paper is justified in Chapter 1 by the fact that NCA was not investigated in other studies so far. It is therefore a bit disappointing, that there are only two sentences related to this method in this section, especially considering that results from the other approaches are described in much greater detail.

p.11,l.17: Isn't it more a change in wind speed that has an effect on rotor speed and tower kinematics alike? Do you mean with this sentence that rotor speed and tower defections are correlated?

p.11,l.22-25: This is surprising. Was it investigated in detail? If there is no correlation, the stepwise regression should not select this feature. Is there an error in the stepwise regression approach or is there another explanation for the pitch angle?

p.11: Was PCA not applied?

p.14, l.16ff: It is not shown in Figure 6 that DELs during standstill are lower as the DELs are normalized. Is there another way to illustrate it?

p.16, l.1: What do you mean by good practice? It seems that feature-selection has no impact on the results. This is contradicting to the research by Sharma and Saroha (p. 6) which states that feature-selection should result in more accurate results. Could you

please discuss?

p.16, table 3: It seems that feature selection has not impact on the estimation results from the NN. Can this indicate that there are still too many features? Usually, each method can be varied by changing some parameters. Have you conducted a parameter study to investigate, if more strict settings for feature selection would result in even smaller feature sets without loosing accuracy?

p.16, l.13f: Please discuss how this result can be used to reduce time and costs for data collection in practice. To my understanding the 40% of the data was randomly selected out of 1 year of measurements. That means that is still requires to measure for one year. To me, one year of measurements sounds reasonable to cover all operational conditions, seasonal variations, etc. I cannot see how that can be reduced really. If still 1 year of measurement is required the purpose of this study remains unclear. Please give a justification why this study was undertaken and why results should be presented in this paper.

p.17,l.11f: I cannot see from table 3 that a NN based on NCA is superior in terms of accuracy. In the abstract it is also mentioned that all NN result in similar accuracy. See also p.15, l. 5ff

Technical comments

p.1, l.17: "with the partial load model"

p.1, l.25: "failures for example"

p.1, l.29: Are there two spaces "of WTs"?

p.1, l.29: "can potentially be"

p.2, l.8: I don't think that "sophisticated" is the correct word. Maybe "challenging"?

p.2, l.10: I don't think that "briefly visited" is the correct wording.

[Figure]

p.2,l.17-34: This section contains quite general statements and also the motivation for performing fatigue load estimation. Why is it placed in the middle of the literature review? Should it be moved to the beginning of chapter 1?

p.3, l.34: I don't think that "scarce" is the correct wording.

p.5, l.13: It is not the method for "development of the paper" which is shown.

p.6, l.11: "most" instead of "more"?

p.6, l.11: "For this purpose"?

p.7, l.23: "differentiate" instead of "differ"?

p.8, l.23: "sought"? What does that mean?

p.8, l.23: "accurately" instead of "appropriately"?

p.8, l.21-23: The same is written just a few lines above.

p.9, l.3-5: This sentence is hard to understand. Could you please formulate it in a different way?

p.12, l.15: "estimated DELs" instead of "trained DELs"?

p.14, l.10: "It can be observed"

---

## Author Comment (AC1) · 13 Apr 2020

Dear Referees,

the authors would like to express their gratitude for the time and effort spend in reviewing our paper. In the attached .pdf document, the response to the referees comments # 1 and # 2 will be given.

Yours sincerely,

On behalve of all the authors,

Marcel Schedat

Enclosures:
- Response to referee comments # 1 (page 2 to 15)
- Response to referee comments # 2 (page 16 to 20)

**Response to referee comments # 1**

Dear anonymous referee # 1,
Thank you very much for your feedback to improve our manuscript!

5 In this document, the authors' responses are added in *cursive*.

As indicated in the contribution, the work presented in this paper originates from the Master's Thesis of the first author. Based on this contribution, this seems to be very good work for a Master's Thesis. In my opinion however, the quality should be improved for this paper. More in particular, especially the presentation quality. Nevertheless, the work presented is interesting

10 and valuable.
Throughout the comments that follow, I indicated the most important (and thus necessary to tackle) ones by (*).

General comments
(*) In general, my main concern is the lack of shown results. Without more information, it is hard to review the results and

15 conclusions drawn. I'll give specific examples later on. A general rule of thumb could be: whenever a conclusion is drawn based on own results, it can only be checked if the results themselves are shown. So all main conclusions drawn in the paper should be preceded by the results shown in a figure or table.

Some other general comments:

20 - The title states that "data mining techniques" (plural) are used to model fatigue loads. However, only neural networks are used. I'd consider to use "neural networks" instead of "data mining techniques". If you want to express the comparison of multiple techniques, an adjustment of the title might be considered towards feature selection (since this is not reflected in the title at this point). In my opinion, this is not strictly necessary.

> *The title was changed to "Feature selection techniques for modelling tower fatigue loads of a wind turbine with*
25 > *neural networks"*

- Although mentioned in the abstract, the sensitivity analysis regarding the length of the data set (and the motivation for it) is not mentioned in the introduction.

> *Sensitivity analysis was changed to a particular reduced set of continues data.*

30
Specific comments
- P.1 line 19-20: it's not clear to me what you mean by "more conservative models regarding the number of features"

> *With the neighborhood component analysis the number of features was reduced while maintaining the interpretability*
> *of the features used. This would not be the case of e.g. Principal Component Analysis.*
35 > *The sentence was changed.*

- P.1 abstract: consider including quantitative results (e.g. errors between measured and estimated DEL do not exceed x%)

> *Quite a lot results were produced for different features and operational modes. We have added absolute mean squared*
> *error for the artificial neural network model using neighborhood component analysis and the full load operational*
40 > *mode.*

- (*) P.4 line 9-10: Some more information on the turbine would be appreciated, in particular rated power.

> *This paper seeks to model tower fatigue loads of a commercial wind turbine with a rated power of 2.05 MW, a hub*
> *height of 100 m and a rotor diameter of 92.5 m in the northern part of Germany. The turbine is used by the Wind*
45 > *Energy Technology Institute at the Flensburg University of Applied Sciences for research purposes.*
> *The text has been changed and added.*

- (*) P.5 line 1: More information about the measurement setup is needed. E.g. Are the measurements corrected for wall temperature before calculating bending moments? Is it possible to show the resulting (normalized) bending moments? E.g. vs time (for a smaller period) and vs windspeed

*For this study, the readings from the SCADA and a load measurement system in the previously mentioned turbine were recorded over around 11 months and collected in 10 min files. The tower bottom bending are measured by strain gauges. These were installed and wired as full bridge (Wheatstone) with temperature compensation. A Wheatstone bridge is widely used in strain gauge applications because of its ability to measure small deviations in resistance. The calibration factors were determined from the results of the shunt-resistor-calibration, tower geometry and the thickness of the tower wall at the strain gauge positions (provided by the turbine manufacturer). The Offsets are determined by means of a yaw round.*

*We have inserted a graph showing the DELs of the tower bending moments vs time (Figure 2).*

*The text has been changed and added.*

- P.5 line 9-11: the number used as n_eq is usually given too

*The short-term damage equivalent loads for every 10 min time series were calculated. The reference number of cycles within the lifetime of 20 years were assumed to be $10^7$ cycles. The short damage equivalent load by 600 s equates to $n_{eq} = 9.5064$. Alternatively, the number of load cycles corresponding to 1 Hz (1Hz DEL) could be possible, but this was not decisive for the focus of the paper.*

*The text has been added.*

- P.5 line 14-16: How are the outliers detected? What were the (normalized) limit values to detect them?

*The process of outlier detection in this study was not automated but done through visual inspection of the descriptive statistics calculated from the time series for each operational mode.*

*The text has been added.*

- P.5 line 19: Can you give some more explanation on the descriptive statistic "mode"?

*"Mode" is one of the seven descriptive statistics that we used. Mode is the most frequent value in a 10-min time series.*

- P.6 Figure 1: nice figure, does help to understand the methodology

*Thank you.*

- p.6 line 4: This seems to be a lot of missing SCADA data. Are all of the variables missing or is it mostly due to one variable?

*As described earlier in the text, measurement errors were removed. In our case this happened due to technical failure in the extraction of the SCADA and our measuring computer over approximately 2.5 months.*

- P.6 line 5: This sentence is a bit confusing. The filtering by operational modes is done for the feature selection and the sensitivity analysis, isn't it?

*Apologies, this is misleading. The filtering was done in general for this analysis, for the sensitivity analysis we used the operational mode of partial load since more data was available there.*

- (*) P.6 line 6-8: should the mean value of ACpow be below or equal to 5kW for the datapoint to correspond to standstill? Or is it the minimum, maximum or another descriptive statistic?

*In this sense, standstill corresponds to 10 min mean "ACpow" readings below or equal to 5 kW (0.25 % of nominal power); partial load to readings higher than 5 kW and below or equal to 2000 kW (97.56 % of nominal power); and full load to readings above 2000 kW.*

*Further explanation has been added.*

- P.6 line 7-8: It is easier to interpret the limits as x% of rated power instead of x kW. Consider to change to relative values.
>*Percentage of nominal power was added (see "- (\*) P.6 line 6-8")*

- P.7 line 7-8: The sentence "Correlation coefficients above 0.95 …" deserves more explanation. Does this mean that if an explanatory feature correlates with more 0.95 to the bending moment it is considered as redundant? Or is this only the case for explanatory features among each other?
>*Correlation above 0.5 between the features and damage equivalent loads was considered. Correlation among all features of a particular sensor above 0.95 was considered as a redundant sensor and therefore eliminated for further analysis.*
>*The text was changed.*

- P.7, line 12: more explanation about p-value and F-statistic would be appreciated.
>*P-value is used as a probability measure to identify if a particular feature is significant for the outcome of the model. If a p-value is larger than 0.05 the null hypothesis is true and the feature is selected for further modelling.*
>*The text was added.*

- P.7 line 22-23 "the output is then predicted by applying a function": What kind of function?
>*The NCA is based on the k-NN algorithm, therefore the prediction is performed by the trained k-NN regression model.*
>*The text has been added.*

- P.7 line 27: how are the weights decided for this paper?
>*The weights are usually (also in the paper) assigned randomly and then adjusted by solving a minimization problem (minimizing the prediction error).*
>*Further text has been added.*

- P.7 line 28: observations = features ? This is a bit confusing, since "observations" might also be used for different measurements (in time)
>*The word was changed to "features".*

- (\*) Section 3.1: a visualization or overview of the selected and disregarded features for each dataset and technique is missing. As a reader, it is impossible to know which features were selected by which technique for which dataset except for the (few) mentioned in the text. Only by showing these results, the drawn conclusions can be checked. Moreover, it's easier to understand the conclusions if you can see the results yourself.
>*We have prepared a detailed overview of the three topics, which strengthens the comprehensibility:*
>*1.  Correlation Analysis by Operation Mode*
>*2.  Stepwise Regression results from all operational modes*
>*3.  Summary of NCA for different operation modes*
>*We think the results are quite interesting but too much too add in the paper. We will therefore write these representations in the appendix A 1 to 3.*

[revised manuscript text omitted]

- P.9 line 19: a figure showing results about the collinearity would be helpful for this discussion
 *All results of the correlating analysis were attached in the appendix A 1.*

- P.9 line 22-23 "Many of the remaining variables … model": an example with specific resulting values would be helpful
 *Same answer as for "P.9 line 19"*

- P.10 line 6: How does stepwise regression avoid multicollinearity?

 *Stepwise regression does not avoid collinearity directly. In Stepwise regression features are added to the regression model one by one. If a feature added results in a better model (meaning that the prediction accuracy is better) then the feature is kept. Generally, a feature which is collinear with others would not improve the accuracy of a model and there it is dropped by the stepwise regression.*

*In order to avoid misunderstandings, we have removed this sentence.*

- P.11 Section 3.1.2: Discussion about PCA seems to be missing

*For each operational mode PCA was performed to account for potential collinearity in the feature-set. This was done consistently with an explained variance of 99 % remaining.*

*The text has been added.*

- (*) P.12 Section 3.2.1: I like the idea to first give more precise results for one neural network model before comparing all of them. However, more information and results should be given. Which were the final features used for this one model? How did the training, testing and validation data look like (for example plot of the measured normalized power curve for all three datasets, can be based on mean statistics)? Plots of the measured and predicted DEL vs mean windspeed, errors vs mean windspeed for example.

*We have rebuilt the entire paragraph to give a better understanding on how precise the results (predicted DELs) in comparison to the measured DELs are. For example, in Figure 5 you can see predicted and the calculated (measured) DELs over a number of 10 min time series and the corresponding prediction error. Furthermore, Figure 6 gives the reader an impression of the behaviour prediction error vs wind speed.*

- P.12 Section 3.2.1: some of the information and results given seem irrelevant for this discussion. E.g. at which epoch the training stopped. Consider to omit this from the paper or to include it in the discussion to show the added value

*We agree with your comment.*

*We removed this part and rebuilt the entire paragraph.*

- P.12 Figure 3: Personally, I don't think this plot adds value to the paper. If not needed for main conclusions, consider to remove it.

*We agree and have therefore removed it.*

- P.12 line 11-13: these datapoints (outliers) are not visible in the figure. Maybe consider a different visualization

*We removed the figure.*

- P.12 line 10-11 and p.13 Figure 4: Personally I don't see the added value of this figure and discussion.

*We removed this part and rebuilt the entire paragraph.*

- (*) P.12 line 12 and further: didn't you exclude outliers from your dataset? Did you take a closer look to the time signals of bending moment and different SCADA parameters to check what is causing these high errors?

*The mean error in percent is calculated as follow:*

$$mean\ error = \frac{(calculated\ DEL - predicted\ DEL)}{calculated\ DEL} * 100$$

*High prediction errors are expected when the DEL is low. For example, the calculated DEL is 30 kNm and the predicted is 150 Nm then we will have an error of -400 %. In terms of prediction error it is extremely high, but we are only 120 kNm off. If we are 120 kNm off in full load then this error can be less then 1 %.*

*We added an additional plot as suggested to see the high error occurring to low wind speed (Figure 6).*

- P.12 line 16: I cannot find the result 0.99486 in the figures or tables

*All models were run again based on better comparability with other studies and the comments made above.*

- P.14 Figure 5: It's not clear to which data sets the figures exactly correspond. Add sublabels please.

*Appendix A 1 to 3 was added for correlation, stepwise regression and NCA for different operational modes.*

*Additionally the whole chapter was rewritten.*

- Section 3.2.1: An additional conclusion could be made: From the introduction I understood Vera-Tudela and Kühn did a similar analysis with slightly different techniques. How do your results compare to theirs?

*Results were changed to "mean error in %" and "standard deviation of the error in %" to make it comparable to the results from Vera-Tudela. The complete analysis was redone therefor changes in numbers are possible.*

*A short comparison was added in Section 3.2.2. in the paper:*

*"The results from the full load model can be compared to existing work from Vera-Tudela and Kühn (2014) where the mean error is below 0.22 for all feature-sets. The maximum absolute error and standard deviation of the error also confirm the results. The accuracy of the results from the partial load model is slightly worse for all features-sets."*

*Additional sentence added in the conclusions: "It can be concluded that the performance of NN is influenced by the operational mode that the WT. The highest accuracy was achieved when the WT was operating in full load and the lowest in stand still."*

- P.14 Section 3.2.2: Additional figures might be helpful here too. For example, measured and predicted (by the different models) DEL vs mean windspeed, where a different color is used for each model/operational mode.

*We added all relevant information on Figure 5 and 6. For all operational modes the calculated DELs vs. predicted DELs highlights the accuracy of the model and the improvement as the WT operates.*

- P.14 Figure 6: if you want to show DELs are lower for standstill than during full load, it is much easier to plot them on the same graph. Moreover, are the results shown here all test data? Why don't you show all test data instead of only 100 points for each operational mode?

*This was intended to highlight that values close to 0 can have a high error in %. The figure 6 was removed and replaced by an overview of the measured DEL vs predicted DEL (figure 5).*

- P.16, Section 3.3: What is the exact intention of this analysis? Is it to determine the minimum period needed to measure the bending moments? If that's the case, shouldn't the focus be on the dataset containing the least data? To make sure a good estimation is obtained for that operational mode too?

*We changed the section by using the first 50 % of the data available in partial load for training the ANN and tested it on to the remaining 50% of the data.*

- P.16 line 11: is the dataset increased consecutive in time or by randomly picking data from the entire dataset of one year?

*No, for the previous sensitivity analysis we also used always the first x percent of data consecutive.*

- Section 3.3: An additional conclusion could be made: From the introduction I understood Smolka and Cheng did a similar analysi. How do your results compare to theirs?

*In the study of Smolka and Cheng only the correlation analysis for selecting features was conducted. In our paper we compared different methods and we were able to reduce the features significantly with NCA. This paper conducted an additional analysis for the standstill mode where significant increase of mean error was identified.*

- (*) p.17 line 11-12: The first part of this conclusion is not clear to me. What do you mean with "conservative model regarding the number of features"? The second part doesn't seem to be true. Looking at Tables 2 and 3, the lowest mean errors are rarely found for NCA.

*The results indicate that using all data and applying neighborhood component analysis for feature selection yields an interpretable and low dimensional feature-set while maintaining high accuracy.*

*The text has been changed.*

- P.18 line 6-10: If the purpose is to eliminate the need for installing strain gauges on every turbine, it seems it is especially necessary the model is validated on a different turbine. Training the model with data from multiple turbines might not be necessary.

>*To be able to generalize the results obtained from this study, the NN model requires validation with data collected from a different wind turbines with the same specifications.*
>*The text has been changed.*

- (*) P.18 line 12: Why would you want to train the neural network with larger datasets? Wasn't one of your conclusions that you didn't need as much data to have a model equally accurate?

>*This is referred to the other operational mode such as stand still and full load. We had limited data there that is why we used partial load to test the model on 50% of the data.*
>*The text has been changed.*

Technical corrections

The comments hereafter are not critical and are meant to improve the readability of the paper.

- p.1 line 12-15: very long sentence, consider to split it up

>*done*

- p.3 line 1-9: different lengths of datasets were used for the different analyses shown in this paper. The summary written here seems to suggest 2,5 months of data was used to model the thrust loads. However, only 2 weeks were used to train the model and (in case of operational data) one year was used to validate. On the other hand, 2,5 months of data was used to perform a Pearson correlation analysis.

>*corrected*

- P.3 line 26-32: If I understood it correctly, your work is actually similar to the work of Seifert et al, except you have done it for tower bending moments, while Seifert et al. did it for blade root bending moments.

>*Our focus was to identify the required feature for build such a model and how or of these effects the accuracy. This was performed by several feature selection techniques. We could conclude, that NCA was able to identify the most relevant features without linear transformation by i.e. PCA while maintaining a high accuracy of the model.*

- P.6 line 4: in my opinion, 6044 hours is not easier to interpret than 36266 observations of 10 minutes. I think "a little over 8 months" would be better. Similarly in the conclusion (p.18 line 3-5)

>*done*

- P.6 line 4: Considered to add also the percentage of remaining data after the removal of missing data.

- P.8 line 7: typo "squared"

>*done*

- P.9 line 25: It might be helpful to clearly state that from this point the second technique, PCA, is discussed.

>*done*

- P.10, Figure 2: add a line at 99% to increase visibility

*You can see the relevant point in Figure 3 (Figure 2 was changed).*

- P.11 line 8: typo "datasets is the again the"
      *corrected*

- P.13 line 5: sentence can be missed very easily
      *The paragraph was completely rewritten.*

- P.14 line 10: reference to results is missing
10      *has been added*

- P.16 line 14: an equivalent number in time (weeks, months) is easier to interpret than 10241 observations
      *The paragraph was completely rewritten and the comments were noted.*

**Response to referee comments #2**

Dear anonymous referee # 2,
We thank you for providing valuable comments to improve our manuscript.
5   In this document, the authors' responses are added in *cursive*.

The paper deals with the estimation of damage equivalent tower bending moments from SCADA data using a neural network
10   approach. In particular it focuses on methods for feature selection to determine which of the available parameters has to be included as input to the network. The work is interesting and worth publication in general. However, improvements are necessary. The paper should also be revised carefully to improve the presentation quality.

General comments
15   The title implies that several data mining techniques for modeling tower fatigue loads are compared. However, it seems that the main work focuses on feature reduction techniques. In my opinion the title should be adjusted to better reflect the content of the paper.

A comprehensive literature survey has been undertaken to investigate state of research related to the estimation of loads from
20   SCADA data. However, the paper also focuses on methods for feature reduction. Are there studies outside of wind energy which compare feature reduction methods? If so, how do the results compare to those presented in this paper?

Specific comments
p.1, l.18-20: What is meant by "conservative" models?
25     *This expression was disadvantageous and was therefore changed to "The Neighborhood component analysis yields the minimum number of features required while maintaining the interpretability with an absolute mean squared error of around 2 % for full load."*
    *In this way, the word "conservative" meant a low dimensional feature-set.*

30   p.1, l.22: What do you mean with "deployment"? It may not be the correct word. How does that relate to the competition?
    *The expression has been replaced by "due to the rapid growth of the wind energy installed capacity".*

p.2,l.7: Could you please explain why load measurement systems are required for this purpose? The structural condition is usually monitored with accelerometers.
35     *For us, the focus was on determining the bending moments on a tubular steel tower of a wind turbine. A continuous measurements and recording of the strain (in this case by strain gauges) were important for this. The acceleration sensors on the tower used in commercial systems are primarily used for vibration monitoring (mostly in the area of the tower head). If a defined limit value is reached, a warning or alarm follows with a corresponding reaction of the control unit (e.g. shutdown). The continuous decoupling of signals from these acceleration sensors was not readily*
40     *available to us at the time of the measurement campaign.*

p.4,Table 1: From the rest of the paper it seems that the dependent variables for the modeling are the fore-aft bending moment and not strains (see p.5, l.11). Please clarify.
    *The tower bending moments are calculated by the measured strains with the help of strain gauges (Chapter 2.1 refers*
45     *to this calculation of the moments)*

p.5, l.2: Please explain what you mean by "short-term" equivalent load?
    *The short-term DEL is based on a 10 min time series without extrapolation to its lifetime. In our chase we used for 10 min $n_{eq} = 9.5064$, which is equivalent to $10^7$ cycles in 20 years. Alternatively, the number of load cycles*
50     *corresponding to 1 Hz (1Hz DEL) could be possible, but this was not decisive for the focus of the paper.*

*The information has been added to the text.*

p.5, l.2: Please explain what you mean by "short-term" equivalent load?
*Repetition of the question before.*

p.5, l.11: Why m=3?
*We determined the DELs on a tubular steel tower. For clarity, we have chosen only one inverse slope m = 3 for steel (m = 4 or 5 are common, too).*

p.7: It is nice that the four approaches are explained briefly. However, it should also be discussed how the approaches compare with respect to feature reduction. What are their limitations or advantages? Maybe the methods can also be compared using a table which may be easier to follow for the reader?
*The results of the methods are now attached to the appendix A 1 to 3 (page 5 to 10 at this document). The reader can see which features were selected for correlation analysis, stepwise regression and NCA.*

p.8, l.8: How does the validation subset generalize the transfer function? And what is meant be "transfer function" anyhow? Is that the network itself?
*The transfer function is part of the ANN. The weights are adjusted and later a weighted sum is passed through the activation function. The validation set is used to "test" the prediction. If the prediction has a bad accuracy but the training data performs well it is overfitted and the weights are readjusted. The expression "transfer function" is changed to "prediction model".*

p.8, l.13: The choice for the NN architecture should be based on some rational and not on a default suggestion by a software. Why do you think that the chosen NN architecture is reasonable? Can you give another justification than that this is the default setting of a software?
*There is no rule on how to choose the hyperparameters for a black box model like ANN. They can be chosen by default or randomly and later adjusted based on the performance of the ANN. In this case there was no need to adjust the parameters as the model performed well. We tried varying the parameters (which is not part of the paper and requires a separate analysis) and got similar results. It could be possible to investigate on the number of hidden layers required to achieve certain results or the learning rates which would be a separate analysis.*

p.8, l.15: See comment above.
*Please see answer above. Our focus was not on optimizing the architecture, which is why we kept the default setting.*

p.9, l.14: The DELs are calculated for the bending moments and not for the thrust.
*Right, the DELs are calculated for the fore-aft moments of the wind turbine tower. We changed the text.*

p.9, l.15: What do you mean by "facing the wind"? Are measurement with large yaw errors included in the dataset? And how can the wind not affect the thrust load?
*In this investigation, we neglected a potential yaw angle error on the system and only calculated the tower bending moment in the fore-aft direction at the base in alignment with the nacelle. The investigation of the influence of inclined wind flows, e.g. due to yaw errors, were not part of this work and is currently under further investigation. The sentence in text has been deleted.*

p.9, l.23ff: Is PCA not applied? Or does the rest of the page refer to PCA?
*The page refers to 11 months of data used to analyse. One of the feature selection techniques is the correlation analysis, where on top PCA is applied for dimensionality reduction. We added this information to the text.*

p.10,l.19-21: The paper is justified in Chapter 1 by the fact that NCA was not investigated in other studies so far. It is therefore a bit disappointing, that there are only two sentences related to this method in this section, especially considering that results from the other approaches are described in much greater detail.

*NCA is described on p. 26 (this document) as a feature selection technique. The technique itself is no particularly novel there for a brief introduction seemed appropriate. We added a complete table on the feature selection techniques and their results. The results from NCA can be found in appendix A 3.*

p.11,l.17: Isn't it more a change in wind speed that has an effect on rotor speed and tower kinematics alike? Do you mean with this sentence that rotor speed and tower defections are correlated?

*The wind speed influences the rotational speed and the rotational speed is correlated with the DEL. If there is no rotation at the wind turbine we still have a changing DEL due to the wind speed and therefore the wind speed is higher correlated with the DEL then only the rotational speed.*
*Here you can see our correlations (appendix A 1):*
- *correlation of mean wind speed (v_wind_mean) with DEL: 0.74*
- *correlation of rotational speed (omega_gen_mean) with DEL: 0.67*
*They do not exclude each other but complement.*

p.11,l.22-25: This is surprising. Was it investigated in detail? If there is no correlation, the stepwise regression should not select this feature. Is there an error in the stepwise regression approach or is there another explanation for the pitch angle?

*If the stepwise regression does not select a particular feature, this does not mean that it is not important or correlated. If a feature does not improve the prediction accuracy if will not be selected by stepwise regression. This can happen when another feature is already selected in a model which provides similar information. In this way multicollinearity is avoided.*

p.11: Was PCA not applied?

*PCA was applied and this information was added to the text.*

p.14, l.16ff: It is not shown in Figure 6 that DELs during standstill are lower as the DELs are normalized. Is there another way to illustrate it?

*This chapter has been completely revised in large parts and the new illustrations provide an improved overview.*

p.16, l.1: What do you mean by good practice? It seems that feature-selection has no impact on the results. This is contradicting to the research by Sharma and Saroha (p.6) which states that feature-selection should result in more accurate results. Could you please discuss?

*Features selection method helped to reduce the amount of information needed to build a particular model. Feature selection reduces the needed information building a particular model since some of the features do not provide any sufficient information to model the DEL. This ultimately leads to a reduced amount of sensors required modelling the DEL (Appendix A1 to 3 added for the feature selection results on correlation analysis, stepwise regression and NCA).*

p.16, table 3: It seems that feature selection has not impact on the estimation results from the NN. Can this indicate that there are still too many features? Usually, each method can be varied by changing some parameters. Have you conducted a parameter study to investigate, if more strict settings for feature selection would result in even smaller feature sets without loosing accuracy?

*This indicates that the feature selection techniques chose the features which has sufficient information to later construct such a model. Investigating the parameters for each feature selection technique is unfortunately not in the scope of this study. It could be possible to reduce the features even further and maintaining the same accuracy, this would need further investigation.*

p.16, l.13f: Please discuss how this result can be used to reduce time and costs for data collection in practice. To my understanding the 40% of the data was randomly selected out of 1 year of measurements. That means that is still requires to

measure for one year. To me, one year of measurements sounds reasonable to cover all operational conditions, seasonal variations, etc. I cannot see how that can be reduced really. If still 1 year of measurement is required the purpose of this study remains unclear. Please give a justification why this study was undertaken and why results should be presented in this paper.

*We changed the sensitivity study to a particular data set (50% of partial load). This were not chosen randomly, they refer to the first 50% of the data. We use the model to predict "future" DELs and investigate how well the model performs. This would mean that collecting around 3 month of data to calculate the DEL would be sufficient to predict it for the next 3 month (for partial load).*

p.17,l.11f: I cannot see from table 3 that a NN based on NCA is superior in terms of accuracy. In the abstract it is also mentioned that all NN result in similar accuracy. See also p.15, l. 5ff

*It is superior in term of least number of features required while keeping the interpretability of the features. This section was completely rewritten.*

Technical comments

p.1, l.17: "with the partial load model"
*corrected*

p.1, l.25: "failures for example"
*corrected*

p.1, l.29: Are there two spaces "of WTs"?
*corrected*

p.1, l.29: "can potentially be"
*corrected*

p.2, l.8: I don't think that "sophisticated" is the correct word. Maybe "challenging"?
*corrected*

p.2, l.10: I don't think that "briefly visited" is the correct wording.

corrected

p.2,l.17-34: This section contains quite general statements and also the motivation for performing fatigue load estimation. Why is it placed in the middle of the literature review? Should it be moved to the beginning of chapter 1?

*corrected*

p.3, l.34: I don't think that "scarce" is the correct wording.
*corrected*

p.5, l.13: It is not the method for "development of the paper" which is shown.
*corrected*

p.6, l.11: "most" instead of "more"?
*corrected*

p.6, l.11: "For this purpose"?
*corrected* to: "For the learning process"

p.7, l.23: "differentiate" instead of "differ"?
*corrected*

5    p.8, l.23: "sought"? What does that mean?
*The Chapter has been revised. The word*

p.8, l.23: "accurately" instead of "appropriately"?
*corrected*

p.8, l.21-23: The same is written just a few lines above.
*corrected*

p.9, l.3-5: This sentence is hard to understand. Could you please formulate it in a different way?

15    *Corrected to: "The results indicate that the accelerations in both directions (i.e. x and y-axis) are highly correlated with the DELs. The standard deviation of the acceleration in the x-direction presents the highest correlation with a coefficient of 0.97, depicting an almost linear relationship between this feature and the dependent variable."*

p.12, l.15: "estimated DELs" instead of "trained DELs"?

20    *Chapter has been revised*

p.14, l.10: "It can be observed"
*corrected*

---

## Referee Report (RR1)

Dear authors,

Thank you very much for the modifications done to your manuscript and taking into account all of my comments. In my opinion it has improved a lot and shows a very interesting application and nice results. Despite the enormous improvement, I still have some (minor) comments left.

- Abstract: At the end of the abstract, you might want to mention you reduced the amount of data by 50% instead of just referring to a reduction.
- Figure 1: there seem to be an error in the graph: shouldn't the most left block say "Eight months" instead of "Partial load"?
- General comment about the results: Although it is a big improvement to include the results, some thought might be giving to the visualisation of them. In my opinion, the tables can be made more compact and therefore easier to compare if the different parameters are given in the columns and the different statistics in the rows for example. Then in each cell, different colours or symbols can be given for different operation modes or techniques. Such a visualisation makes it possible to easily deduct from the tables which parameters (disregarding the statistics) or which statistics (disregarding the parameters) are most important.
- P. 9 line 8: you depict the maximum and mean windspeed as being highly correlated, which is true. But this statements makes the reader think these are the only two statistics of windspeed that are high correlated. Which is not true, because range and std seem to be even higher correlation than the mean value. Consider to rephrase the sentence. E.g. Several statistics of wind speed, such as the mean value, and power are also highly correlated.
- P. 9, Figure 2: It is a bit confusing both graphs don't show the same time interval. Moreover, I suppose the purpose of this figure is to illustrate the correlation between (mean) wind speed and measured DEL. In my opinion, this is not clear from the graph. A graph of Measured DEL vs mean windspeed would make this better visible I suppose. Another option is to zoom in on a period (of the same length for both) where this correlation is very obvious.
- P. 9: I'm missing a discussion about the results for pitch angle, since this a parameter that is generally important. So even though the results for pitch angle are not high, it might be useful to point that out and explain.
- P. 9, line 19-...: The placement of this paragraph is a bit confusing. You start the section with 56 features and all of a sudden you have more features (63) at a certain point while you expect a decreasing number for features. Consider to move this paragraph (or at a least a part of it) to the beginning of the section.
- P. 10, line 5-6: Although correct, the compactness of the sentence "After removing ... included in the model." might be difficult to understand. Splitting it up into the removal of features due to Pearson correlation and the combining of the remaining features due to PCA might help.
- P. 11, line 19: I think there might be an error and it should be "mean generator speed" instead of "mean wind speed".
- P. 13, Figure 4: the caption might be adjusted a bit with some information on which features were used for the model giving these results. E.g. the resulting features when applying Pearson correlation on the entire dataset.
- P. 14, Figure 5: typo in the caption: "bottom" instead of "button"

- P. 16, line 6-8: It is not clear to me what is meant with "The maximum absolute error … confirm the results". Which results are confirmed? Those of Vera-Tudela and Kühn? How?
- P. 16 line 13: I think something went wrong here: half a sentence is written and there seem to be a title of a section missing. If this is not a mistake, I do think it is better to have a separate section on the reduced data set for modelling.
- P. 17, line 6: "from the model trained with the remaining data". This part seems to indicate two models were trained, one with the first part of the data and one with the second part. Is this the case? Because the remainder of the paper seem to indicate this is not the case.

---

## Author Response (AR2)

Dear Referees,

thank you very much for reviewing our paper.
Please find attached our response to your comments.

Yours sincerely,

On behalve of all the authors,

10    Marcel Schedat

Enclosures:
- Response to referee comments # 1 (page 2 to 3)
15    - Response to referee comments # 2 (page 4 to 6)
- detailed overview of the changes made to the paper (page 7 to 42)

**Response to referee comments # 1**

Dear anonymous referee # 1,
Thank you very much for your feedback to improve our manuscript!
5    In this document, the authors' responses are added in *cursive*.

Abstract: At the end of the abstract, you might want to mention you reduced the amount of data by 50% instead of just referring to a reduction.
    *The abstract has been updated to include the 50 % reduction.*

Figure 1: there seem to be an error in the graph: shouldn't the most left block say "Eight months" instead of "Partial load"?
    *Figure 1 has been updated.*

General comment about the results: Although it is a big improvement to include the results, some thought might be giving to
15    the visualisation of them. In my opinion, the tables can be made more compact and therefore easier to compare if the different parameters are given in the columns and the different statistics in the rows for example. Then in each cell, different colours or symbols can be given for different operation modes or techniques. Such a visualisation makes it possible to easily deduct from the tables which parameters (disregarding the statistics) or which statistics (disregarding the parameters) are most important.
20    *The table A 1 has been updated and merged into a single table for a better overview of the features selected for each operational mode.*

P. 9 line 8: you depict the maximum and mean windspeed as being highly correlated, which is true. But this statements makes the reader think these are the only two statistics of windspeed that are high correlated. Which is not true, because
25    range and std seem to be even higher correlation than the mean value. Consider to rephrase the sentence. E.g. Several statistics of wind speed, such as the mean value, and power are also highly correlated.
    *The sentence has been rephrased.*

P. 9, Figure 2: It is a bit confusing both graphs don't show the same time interval. Moreover, I suppose the purpose of this
30    figure is to illustrate the correlation between (mean) wind speed and measured DEL. In my opinion, this is not clear from the graph. A graph of Measured DEL vs mean windspeed would make this better visible I suppose. Another option is to zoom in on a period (of the same length for both) where this correlation is very obvious.
    *Figure 2 has been updated to reflect the same time interval. An additional figure (Figure 2 (c)) plotting mean wind speed vs. measured DELs in the three operational modes has been added.*

P. 9: I'm missing a discussion about the results for pitch angle, since this a parameter that is generally important. So even though the results for pitch angle are not high, it might be useful to point that out and explain.
    *The pitch angle is held at the most efficient lift-to-drag ratio during the partial load and, therefore, not many variations can be observed during standstill and partial load. During full load, the turbine pitches constantly*
40    *continuously to keep the rotational speed nearly constant.*
    *We updated the text: "Relationships between sensor signals and the estimated DELs can vary depending on the operational mode of the wind turbine i.e. the pitch angle operates mainly during startup and full load. Methods with an underlying linear assumption, such as the correlation analysis, can lead to misinterpretation of feature importance when observing the complete dataset."*
45

P. 9, line 19-...: The placement of this paragraph is a bit confusing. You start the section with 56 features and all of a sudden you have more features (63) at a certain point while you expect a decreasing number for features. Consider to move this paragraph (or a least a part of it) to the beginning of the section.
    *This has been clarified and section 3.1.1 has been updated.*
50

P. 10, line 5-6: Although correct, the compactness of the sentence "After removing ... included in the model." might be difficult to understand. Splitting it up into the removal of features due to Pearson correlation and the combining of the remaining features due to PCA might help.

*This has been clarified and updated.*

P. 11, line 19: I think there might be an error and it should be "mean generator speed" instead of "mean wind speed".

*This was a mistake. We have changed the text to mean generator speed.*

P. 13, Figure 4: the caption might be adjusted a bit with some information on which features were used for the model giving these results. E.g. the resulting features when applying Pearson correlation on the entire dataset.

*The caption has been updated.*

P. 14, Figure 5: typo in the caption: "bottom" instead of "button"

*The caption has been corrected.*

P. 16, line 6-8: It is not clear to me what is meant with "The maximum absolute error ... confirm the results". Which results are confirmed? Those of Vera-Tudela and Kühn? How?

*This might have been misleading, we wanted to point out that the results are comparable in general. By saying "confirming the results" we wanted to highlight that they are in similar ranges. We can see now why this is misleading and we have rephrased the sentence.*

P. 16 line 13: I think something went wrong here: half a sentence is written and there seem to be a title of a section missing. If this is not a mistake, I do think it is better to have a separate section on the reduced data set for modelling.

*It was a formatting mistake and is a new subsection.*

P. 17, line 6: "from the model trained with the remaining data". This part seems to indicate two models were trained, one with the first part of the data and one with the second part. Is this the case? Because the remainder of the paper seem to indicate this is not the case.

*We see now the misunderstanding. We were trying to explain, that we have the model trained with the first 50 % of the data and are predicting the remaining 50 % of the data (how we would consider doing it in a continuous monitoring system). We have rephrased the sentence.*

Dear anonymous referee # 2,
We thank you for providing valuable comments to improve our manuscript.
5 In this document, the authors' responses are added in *cursive*.

General comments

10 1) As stated in the paper, the novel aspects of the paper, for example in relation to work carried out by Vera-Tudela et. al., is the application of NCA. However, this is not sufficiently reflected in the paper in my opinion. It seems to focus on reproduction of results from other studies instead. For example, the paper does not use the opportunity to compare strength and weakness of NCA against the other approaches applied. In addition, interpretation of results from NCA is missing while results from correlation analysis are discussed in detail for example. I addressed this discrepancy between justification of the
15 paper and the actual content already in my last review, i.e. comment p.7, comment p.10, l.19-20 and general comment on literature survey which were not taken into account.

  *In this round of comments, we have tried to highlight NCA as the novelty of the paper and have elaborated on the results and performance of this technique in more detail. The discussion of the results from the complete dataset has been expanded to include the following text:*
20 - *"NCA did not select features such as the variance of the acceleration in y-direction and the minimum windspeed during a 10 min timeseries that had been selected by the correlation analysis and stepwise regression. This shows that to model the DELs with NN, these features are not relevant and can be omitted without compromising the model's performance as suggested by the mean error of 2.07 % in Table 2."*

  *The discussion of the results for the different operational modes has been updated to include the following:*
25 - *"NCA outperformed all other methods in terms of mean error in standstill and full load (see Table 4). In partial load, NCA still performed well, however, correlation and correlation & PCA yielded slightly lower mean errors (see Table 4)."*

  *Additionally, we have added a table to compare the methods and highlight their corresponding strengths and limitations.*

2) I also made a general comment regarding presentation quality. I still find that the language is imprecise at times and some of my comments in the last review addressed such issues. However, this was not taken sufficiently into account and some of my comments below still relate to imprecise language. In that respect my comments below are not comprehensive and I recommend to carefully revise the paper in this respect.
35   *We have revised a large number of statements and significantly reduced the linguistic inaccuracy in the text.*

Specific comments:

page 20, line 30:
40 Rapid growth is not an explanation for a fierce competition in my opinion. Maybe it could be stated instead that there is still the need to reduce the cost of wind energy further and that improved monitoring solutions can potentially decrease O&M costs?

  *Agree, this paragraph has been rephrased.*
  *"To ensure the cost-competitiveness of this technology in the future, it is important to seize the potential cost*
45  *reductions related to operation and maintenance (O&M). This includes improving monitoring solutions and life extension strategies."*

p.24, Table 1:
Thanks for your answer. Shouldn't the description of the dependent variables then read "Bending moment derived from
50 gauge sensor located..." instead of "Gauge sensor located...", especially as the unit is given as kNm?

*The definition has been corrected.*

p. 24, line 10:
I suggest to write "This calculation..." instead of "This transformation..." because in the sentence before "transformation" refers to deriving bending moments from strains.
*We agree, it has been updated.*

p. 26, line 6:
Is the reason to split the data set into partial and full load operation also that some of the applied methods (Regression, PCA) are linear methods, i.e. cannot handle non-linear relations? If so, this could be mentioned either here or in the section where the methods are explained briefly.
*Updated to: "Relationships between sensor signals and the estimated DELs can vary depending on the operational mode of the wind turbine i.e. the pitch angle operates mainly during startup and full load. Methods with an underlying linear assumption, such as the correlation analysis, can lead to misinterpretation of feature importance when observing the complete dataset."*

p. 28, line 20:
Thanks for your explanation. I agree that there is no general rule how to chose the network topology. Because of that, testing different network topologies is usually required important. I understand from your reply that this was done at least regarding the number of neurons in the single hidden layer and that can be mentioned in the text. If I misunderstood you reply, you should at least justify the choice of the network by the satisfying prediction accuracy that you experienced during testing. Currently this sections reads as if you did not care about the network topology which according to your reply was not the case.
*Yes, we have tried different amount of neurons but the performance has not improved. We have now mentioned it in the text and included a reference to a similar study where 25 neurons were chosen to model tower fatigue loads by Lind et al. (2017). We had initially used this study as reference for our configuration. We have updated the text to include the following:*

*"The NN was initially set to 25 neurons in the hidden layer and 1 neuron in the output layer as per Lind et al. (2017). However, we tested different configurations and found that the results remain consistent. Therefore, the number of neurons in the hidden layer are set to 10 neurons and 1 neuron in the output layer. This simple configuration reduces the computational complexity and time while enabling the modelling of non-linear relationships."*

p. 30, line 21:
In the paper the term 'feature' relates to an explanatory variable. I recommend to replace the word 'feature' by 'principal component' in this sentence to distinguish between these two.
*Has been changed.*

p. 32, line 21:
Thanks for your explanation. I still think that the sentence is imprecise. What do yo mean by deviation? Do you mean a change of wind speed? Also the next sentence does not make sense.
*Has been rephrased.*
*"For example, fluctuations of the rotational speed during full load have a significant effect on the tower movement which explains the high correlation between the standard deviations of the rotational speed with the DELs.*
*Additionally, several descriptive statistics of the pitch angle are correlated with the DEL exhibiting correlation coefficients greater than 0.5."*

p. 32, line 30:

Thanks for your reply. However, it does not address my comment. I was wondering why the pitch angle was selected by stepwise regression at all, as it does not correlate with the DEL. Same holds true for air density an wind direction. Please clarify.

*By the nature of stepwise regression a feature addet to the stepwise regression process has to improve the overall accuracy which highly depends on the previous features added already. If a particular feature does not improve the overall accuracy then it will be dropped. The method is not very reliable, also it is biast twards the previous features which have been added to the model. We have created a table summerizing the pros and cons of each feature selection / dimention reduction technique.*

10 p.34, line 3:
Do you refer to Figure 4 or Figure 5 here?

*Yes, we meant Figure 5. We have corrected this.*

p.34, line 6:
15 A few lines above you already mentioned that normalized predicted vs measured DELs is shown in Figure 5.

*We agree and have removed the redundant sentence.*

p. 35, line 5:
It seems that the turbine is in operation also at wind speeds lower than 5 m/s. Units should be corrected from m/s^2 to m/s.

*It is a wind speed of 3 m s⁻¹. We have confused it with the power of 5 kW under which the wind turbine is in standstill (see our definitions for standstill, partial load, and full load).*

p. 35, line 9:
What to you mean by 'significantly higher results'?

*We have included the ranges of variations of the mean error.*

Appendix 1-3:
Many thanks for this additional information. For a better overview and comparison of the methods there could be one table only, where a cell contains "0.9, b, c" if this variable was used by all approaches and "0.9, c" if it was used by correlation and
30 NCA for example. In that way only one table is needed and it could maybe be integrated into the main body of the paper (not Annex).
Please also change 'still stand' to ' stand still' for consistency with the rest of the paper.

*It is updated and merged into one table.*

[revised manuscript text omitted]